# OmniArch: Building Foundation Model For Scientific Computing

**Tianyu Chen**[1]   **Haoyi Zhou**[1]   **Ying Li**[2]   **Hao Wang**[2]   **Chonghan Gao**[1]   **Rongye Shi**[3]
**Shanghang Zhang**[2]   **Jianxin Li**[1]

## Abstract

Foundation models have revolutionized language modeling, while whether this success is replicated in scientific computing remains unexplored. We present OmniArch, the first prototype aiming at solving multi-scale and multi-physics scientific computing problems with physical alignment. We addressed all three challenges with one unified architecture. Its pre-training stage contains a Fourier Encoder-decoder fading out the disharmony across separated dimensions and a Transformer backbone integrating quantities through temporal dynamics, and the novel PDE-Aligner performs physics-informed fine-tuning under flexible conditions. As far as we know, we first conduct 1D-2D-3D united pre-training on the PDEBench, and it sets not only new performance benchmarks for 1D, 2D, and 3D PDEs but also demonstrates exceptional adaptability to new physics via in-context and zero-shot learning approaches, which supports realistic engineering applications and foresight physics discovery.

## 1. Introduction

Developing robust neural surrogate models for temporal partial differential equations (PDEs) is crucial for various scientific and engineering applications, including aircraft design, weather forecasting, and semiconductor manufacturing (Allen et al., 2022; Pathak et al., 2022). These PDEs describe spatial-temporal dynamic systems that are foundational to these industries. Traditional scientific computing methods, such as Finite Element Methods (FEMs) and Finite Volume Methods (FVMs) (Oden, 1989), require extensive handcrafted coding and are computationally intensive, even

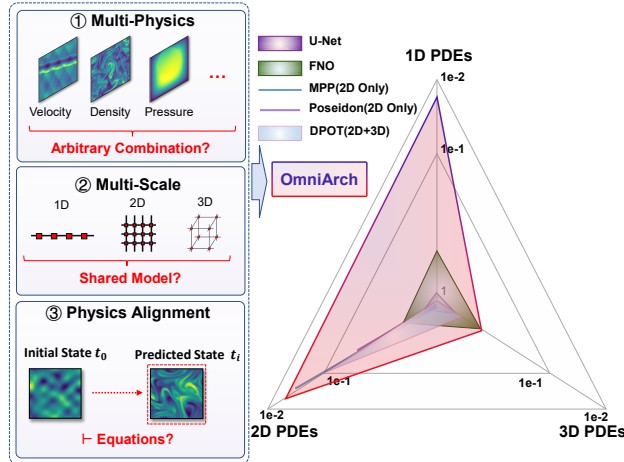

Figure 1: OmniArch achieves state-of-the-art performance (nRMSE Loss) on 1D-2D-3D PDE tasks with single foundation model. The baselines include the task-specific expert models and the pre-trained models.

on state-of-the-art High-Performance Computing (HPC) clusters. To expedite PDE solving, pioneers have explored the construction of neural operators that learn mappings between function spaces, offering the potential to generalize across different discretizations. For the requisite precision, neural operators are often enhanced with physics-informed normalization techniques, such as customized loss functions derived from the governing physical equations (Raissi et al., 2019).

The primary limitation of neural operator methods lies in their case-specific design, restricting their application scope and hindering broad transferability across diverse physical systems. Recent efforts aim to enhance the transferability of neural operators by developing foundational models that leverage advancements in learning strategies, architectural design, and data curation (Alkin et al., 2024; Sun et al., 2024; Shen et al.). In terms of learning, the *pre-train and fine-tune* paradigm, proven effective for Fourier Neural Operator (FNO) models (Subramanian et al., 2023), has been adapted to PDE contexts. Additionally, Lie group-based self-supervised learning (Lie-SSL) (Mialon et al., 2023) introduces physics-constrained transformations for PDEs, primarily addressing inverse problems. Architec-

[1]SKLCCSE, School of Computer Science and Engineering, Beihang University, Beijing, China [2]SKLMIP, School of Computer Science, Peking University, Beijing, China [3]School of Artificial Intelligence, Beihang University, Beijing, China. Correspondence to: Jianxin Li <lijx@buaa.edu.cn>.

*Proceedings of the $42^{nd}$ International Conference on Machine Learning*, Vancouver, Canada. PMLR 267, 2025. Copyright 2025 by the author(s).

turally, innovations like ICON_LM (Yang et al., 2023b), and PITT (Lorsung et al., 2023) incorporate language model principles to enhance neural operator learning, enabling generalization through equation captions. The Factformer (Li et al., 2023) introduces a scalable transformer for multi-dimensional PDE data, with the Multi-Physics Pre-training (MPP) (McCabe et al., 2023), Poseidon (Herde et al., 2024) and DPOT (Hao et al., 2024) further extending this approach to 2D data pre-training. From a data-centric viewpoint, resources such as PDEBench (Takamoto et al., 2022), PDEArena (Gupta & Brandstetter, 2022) and The-Well (Ohana et al., 2024) offer well-structured datasets that facilitate pre-training and the establishment of rigorous benchmarks.

While attempting unified learning of multiple PDE solvers in a single model, multi-scale and multi-physics challenges persist. The above surrogate models, often constrained by the fixed mapping grid (MPP, Lie-SSL, ICON_LM) and single-time step observation window (MPP, Factformer, PITT, Poseidon), struggle with flexible spatial grid input and long-sequence roll-out predictions.

In this work, we study how to frame the foundation model learning paradigms for Scientific Computing tasks w.r.t PDEs, namely OmniArch. For the pre-training stage, we define a flexible pipeline to deal with multiple-physics spatial-temporal data and convert the forward problem learning into popular auto-regressive tasks that can be scaled up easily. For the pre-training stage, we devise a flexible pipeline to handle multi-physics spatio-temporal data and reformulate the forward problem as scalable autoregressive tasks. Specifically, we employ a Fourier encoder to convert coordinate and observation data into frequency components (modes). We use truncated modes to form PDE token embeddings, sequenced for processing by transformer blocks, and we design the PDE-Aligner during fine-tuning to align predictions with known physical laws and principles, improving the model concordance to conventional physical constraints.

We release our models' base and large variants[1], concurrently addressing 1D, 2D, and 3D PDEs. Evaluating performance across 11 PDE types from PDEBench and PDEArena, our OmniArch achieves state-of-the-art results, as illustrated in Figure 1. For the Computational fluid dynamics (CFD) related tasks, we observe one to two orders of magnitude reductions in normalized root mean squared error. Moreover, our models exhibit emergent capabilities, such as zero-shot generalization to novel PDE systems and in-context learning of neural operators. The representations learned by OmniArch demonstrate versatility, readily adaptable to inverse problems. Notably, OmniArch facilitates multi-scale inference, accommodating a range of input grid resolutions with moderate precision trade-offs. In summary, our key

contributions and findings include:

- We introduce OmniArch, the first foundation model to successfully conduct 1D-2D-3D united pre-training. Using a Fourier Encoder-decoder, OmniArch allows for flexible grid inputs, enabling unified multi-scale training. The Temporal Mask effectively addresses inconsistencies in multi-physics systems, allowing different physical quantities and time steps to be learned simultaneously within a shared Transformer backbone.

- We develop the PDE-Aligner for physics-informed fine-tuning, which leverages hidden representations of equations and other physical priors to align with observed physical field dynamics.

- After fine-tuning, OmniArch achieves state-of-the-art performance on 11 types of PDEs from the PDEBench and PDEArena benchmarks. The model exhibits in-context learning capabilities and demonstrates promising zero-shot performance.

## 2. Related Works

**Learned PDE Solvers.** Deep Learning for solving PDEs has been a recent focal point of research (Lu et al., 2021b; Karniadakis et al., 2021), including physics-informed methods (Raissi et al., 2019), GNN-based techniques (Veličković et al., 2017; Pfaff et al., 2020), and neural operator models like DeepONet (Lu et al., 2021a) and FNO (Li et al., 2020). While effective, these models often require task-specific training and struggle with generalization. ICON_LM (Yang et al., 2023a), MPP (McCabe et al., 2023), PDEformer-1 (Ye et al., 2024) aim to generalize across diverse physical systems but limit to a single dimension.

**Foundation Models for Science.** The Foundation Models (Devlin et al., 2018; Brown et al., 2020; Radford et al., 2019; 2018; Touvron et al., 2023; Radford et al., 2021) have emerged as pivotal elements in the field of natural language processing, computer vision, and cross-modal tasks. After large-scale pre-trained with the transformer backbone, they serve as the bedrock for a multitude of downstream tasks by fine-tuning (Zhang et al., 2023) or in-context learning (Li, 2023). Recently, they have shown promise in scientific fields, exemplified by FourcastNet (Pathak et al., 2022) for weather forecasting, OpenLAM (Zhang et al., 2022) for chemistry, and HyenaDNA (Nguyen et al., 2023) for biomedical tasks. However, applying foundation models to scientific computing, particularly PDE solving, remains an emerging and pioneering area.

## 3. Method

The foundation models (Devlin et al., 2018; Brown et al., 2020; Radford et al., 2019; 2018; Touvron et al., 2023;

---

[1] https://openi.pcl.ac.cn/cty315/OmniArch

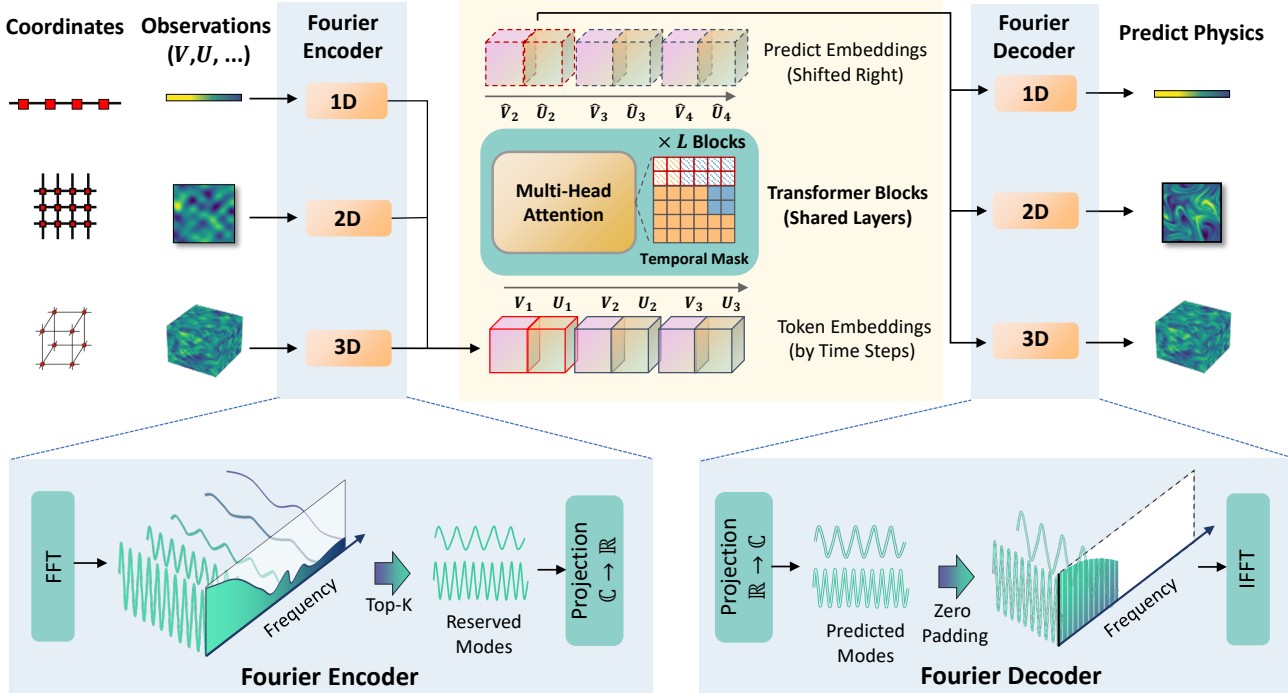

Figure 2: **The overview of OmniArch.** The *Fourier Encoder* converts coordinates and physical fields into frequency domains, enabling unified training for 1D, 2D, and 3D data. Reserved frequency modes form PDE token embeddings for *Shared Transformer Blocks*. Tokens are grouped by timestep to create a *Temporal Mask* for prediction. Predicted modes are decoded using IFFT with zero padding to recover the physical field.

Radford et al., 2021) have shown significant success with broad generation to various inputs and downstream tasks. Building a similar model for scientific computing should require addressing dynamic and complex physical systems and learning intrinsic laws from wild physical phenomena. We highlight the major challenges as three-fold:

**Multi-Scale** The ability to handle inputs of different dimensions (1D, 2D, 3D), varying grid resolutions, and diverse grid shapes. For example, fluid dynamics simulations can range from simple one-dimensional pipe flow to complex three-dimensional turbulent flow, and the model must maintain accuracy and consistency across these different scales.

**Multi-Physics** The capability to handle dynamic systems involving different physical quantities. For instance, in meteorology, multiple physical quantities such as wind speed, temperature, and humidity interact, requiring the model to process these different physical fields simultaneously.

**Physical Alignment** Allowing flexible incorporation of physical priors such as governing equations, symmetries, conservation laws, and boundary conditions into the solution process. For example, in heat conduction problems, the law of conservation of energy and boundary conditions is crucial for predicting temperature distributions.

The proposed OmniArch Model follows the predominant *pre-training-then-fine-tune* paradigm. In subsection 3.1, we utilize Fourier Encoders and Decoders to address the multi-scale challenge and employ the Temporal Attention mechanism to handle multi-physics generalization problems. In subsection 3.2, we leverage the PDE-Aligner in the fine-tuning stage, allowing the incorporation of physical priors in textual form into the model's learning and adaptation process.

### 3.1. Pre-training OmniArch: Flexibly Learning from Different Dynamic Systems

The overall pre-training framework of OmniArch is illustrated in Figure 2. For physical data of different dimensions (1D, 2D, 3D), we use separate Fourier Encoders to transform their coordinates and observed physical field values into the frequency domain. High and low frequencies are truncated in the frequency domain so that data from different grids have the same length of embedded representations. Then, these representations are processed through shared Transformer modules to model the integral operators along the time axis. We leverage the Temporal Mask to ensure that each physical quantity can simultaneously attend to all physical quantities and previous time steps. Finally, the predicted embedding representations are used to recover

the predicted frequency domain signals. We involve zero-padding to keep these signals with the target physical field shape and perform individual inverse Fourier transforms to output the corresponding physical field predictions.

### 3.1.1. ENCODER/DECODER IN FOURIER DOMAIN

The multi-scale challenge needs proper representation of inputs from different dimensions, varying grid resolutions, and shapes. Inspired by the Fourier transforms (Brigham, 1988) convert the sequential signals into frequency components, we re-organize the multi-scale inputs in the spatial domain into the multi-component ones in the frequency domain. The traditional pipeline includes convolutional encoders (Raonic et al., 2023), which capture the local features in separated dimensions while the global information exchange happens at the channels' explicit mixing. The results of Fourier transforms are complex coefficients that measure the magnitude and phase of decomposed periodic components and the global information is naturally weighted, which also applies to the complex boundary conditions and heterogeneous grids. Based on that, we further introduce the filter-like components selecting mechanism that distinguishes the high-frequency (detailed variations) and low-frequency (overall trends) ones in physical inputs, which may maintain different patterns and distribution ratios among the local and global representation. Thus, we can build a universal representation with different resolutions and grid shapes in one flexible network architecture.

From a computing-efficient perspective, the forward procedure of Fourier Encoders can be implemented through the Fast Fourier Transform (FFT) with the $O(N \log N)$ complexity while the convolution operation ends in $O(N^2)$. The sparsity and separability of frequency domain features facilitate the subsequent Transformer modules in efficiently processing temporal information, reducing the model's parameters and computational overhead for better training and inference efficiency.

Let $\mathcal{U} \in \mathbb{R}^{T \times D \times 1}$ stand for the physical field inputs. If we have a real-valued input $u(x^{(d)}, t) \in \mathbb{R}$ from $d$-th index and $t$-th time step, the Fourier Encoder firstly applies FFT to convert it from the spatial domain to the frequency domain. Note that $D$ is the total dimension and $d$ denotes the sequential index $(1, 2, 3 \ldots)$, for example, $D = D_1 + D_2 + D_3 = 6$ for 1D, 2D and 3D inputs. Then we have the frequency domain representation $\hat{\mathcal{U}} \in \mathbb{C}^{T \times F \times 1}$ after traversing through all time steps and dimensions. As previously discussed, we design a filter-like mechanism by applying the TopK selection on all $F$ components (modes) in the frequency domain. For the $t$-th time step, all the $K$ significant $(K < F)$ components $\hat{u}_K(t)$ are retained and form the truncated frequency domain. To be clarifying, we can present the forward proce-

dure of $k$-th largest components $\hat{u}_K(k, t)$ as:

$$\hat{u}_K(k, t) = \text{TopK}(\text{FFT}(\, \mathbf{\Psi}[u(x^{(1)}, t), \ldots, u(x^{(D)}, t)]^\top \,)),$$
(1)

where $\text{TopK}(\cdot)$ denotes the selection operator over $F$ components, $\mathbf{\Psi}(\cdot)$ denotes the linear projection for the dimension alignment and the $\text{FFT}(\cdot)$ operator is performed at the individual time step.

In the decoding stage, the predicted frequency domain features $\hat{u}_K^{\text{pred}}(k, t)$ are adapted to the target shape using zero padding. Then, the inverse Fourier transform (IFFT) is applied to revert the frequency domain features $\hat{u}^{\text{pred}}(k, t)$ back to the spatial domain, ultimately obtaining the predicted physical field $u^{\text{pred}}(x^{(d)}, t+1)$ as:

$$u^{\text{pred}}(x^{(d)}, t+1) =$$
$$\mathbf{\Psi}'(\, \text{IFFT}(\, \text{Zero-Padding}(\, [\hat{u}_K^{\text{pred}}(1, t), \ldots, \hat{u}_K^{\text{pred}}(K, t)] \,)) \,).$$
(2)

This encoding and decoding process in the frequency domain is maintained throughout the whole OmniArch network. Since the encoding and decoding operations are always conducted along specific dimensions, thus we omit the $d$-th index indicator in the following context.

### 3.1.2. TRANSFORMER AS AN INTEGRAL NEURAL OPERATOR

To achieve multi-physics versatility, we leverage the Transformer backbone to simulate integral neural operators. In physics, multi-physics systems often exhibit complex spatio-temporal dependencies, requiring effective long-range dependency modeling. The multi-head self-attention mechanism of the Transformer, with the introduction of the Temporal Mask, allows each time step to attend to all physical quantities at the same and previous time steps, enabling efficient temporal information integration. This design ensures the robustness and adaptability of the model in multi-physics systems. Additionally, by padding variable-length sequences, systems with different numbers of physical quantities can use the model for temporal regression predictions in batches, ensuring accuracy and stability.

Moreover, the autoregressive mechanism of the Transformer bears a strong mathematical resemblance to traditional multi-step methods for solving equations. Traditional multi-step methods approximate solutions iteratively, capturing the dynamic changes of the system. Similarly, the multi-head self-attention mechanism of the Transformer models the global dependencies at each time step, achieving precise capture of dynamic changes in the system.

Specifically, traditional multi-step methods for solving equa-

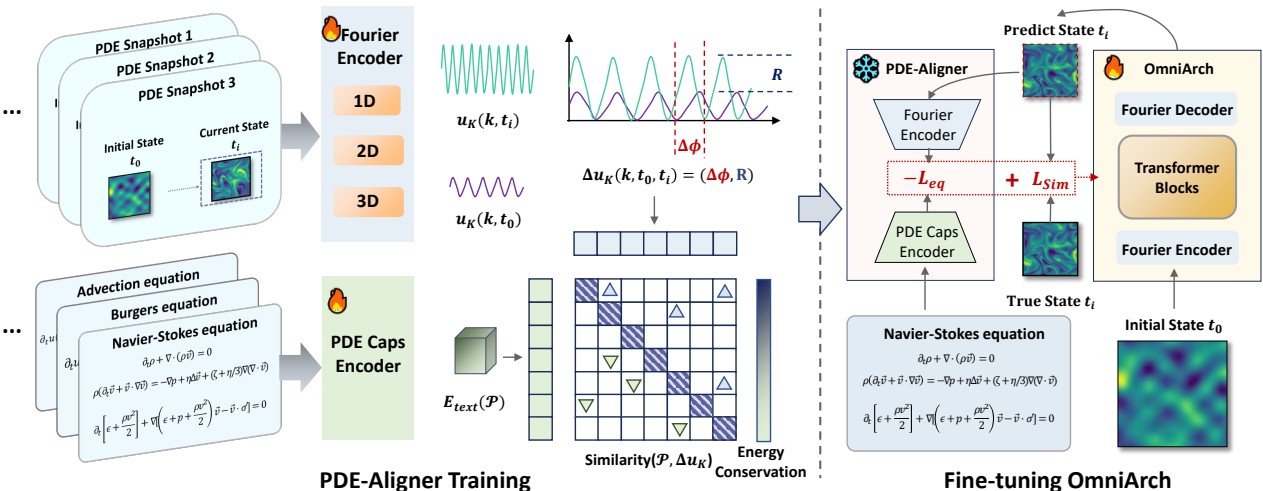

Figure 3: **(Left) PDE-Aligner** architecture with *Fourier Encoders* for initial/current state, and *PDE Caps Encoder* enforcing consistency via PDE constraints. **(Right) Fine-tuning OmniArch with PDE-Aligner** on downstream PDEs like Navier-Stokes equations for physics-informed learning.

tions can be expressed iteratively as:

$$u^{\text{pred}}(x, t+1) = u(x, t) + \Delta t \cdot f(u(x, t)). \quad (3)$$

In contrast, the autoregressive mechanism of the Transformer updates the current state by a weighted sum of previous time steps through attention weights:

$$\begin{aligned} u^{\text{pred}}(x, t+1) &= \Sigma_{i=1}^{t} \alpha_{i,t} \cdot u(x, i) \\ &= u(x, t) + \Sigma_{i=1}^{t-1} \alpha_{i,t-1} u(x, i), \end{aligned} \quad (4)$$

where $\alpha_{i,t}$ refer to the attention weights. Both approaches update based on previous time steps, with the attention mechanism acting as a neural surrogate (Sun et al., 2020) for the integral operator $f$.

Assume that we have a physical system with two physical quantities $u(x, t)$ and $v(x, t)$, where the total number of quantities is recorded by $C = 2$. In OmniArch's computation, the frequency domain features $\hat{u}_K(k, t)$ and $\hat{v}_K(k, t)$ obtained from the Fourier Encoder are further transformed into real-valued embeddings through $\mathcal{R}(\cdot)$, resulting in the input embeddings for the Transformer $\mathbf{U}_t$ and $\mathbf{V}_t$. These embeddings are grouped by time steps to form the input sequence $\mathbf{Z}_t$. For each time step $t$,

$$\mathbf{Z}_t = \{\mathbf{U}_t, \mathbf{V}_t\} = \{\mathcal{R}(\hat{u}_K(k, t)), \mathcal{R}(\hat{v}_K(k, t))\}. \quad (5)$$

The Temporal Mask $\mathbf{M}$ ensures that each time step $t$ can access all physical quantities at the current and previous time steps, which is defined as:

$$\mathbf{M}(i, j) = \begin{cases} 0 & \text{if } \lfloor \frac{j}{C} \rfloor \leq \lfloor \frac{i}{C} \rfloor \\ -\infty & \text{if } \lfloor \frac{j}{C} \rfloor > \lfloor \frac{i}{C} \rfloor \end{cases}, \quad (6)$$

where $i$ and $j$ represent the $i$-th and $j$-th tokens in the sequence, and $\lfloor \frac{i}{C} \rfloor$ represents the time step. Unlike standard causal masking that enforces strict sequential dependencies, our Temporal Mask enables all physical quantities within the same timestep to attend to each other, addressing the fundamental coupling inherent in multi-physics systems. Specifically, for a system with $C$ physical quantities at each timestep, tokens $\{i, i+1, ..., i+C-1\}$ corresponding to timestep $t$ have full visibility of each other (intra-timestep attention), while maintaining causal relationships across timesteps (inter-timestep attention). This hierarchical attention pattern ensures that coupled physical quantities—such as velocity and pressure in fluid dynamics—can jointly evolve while respecting temporal causality. The design is particularly crucial for systems where physical variables must satisfy simultaneous constraints (e.g., continuity equations in Navier-Stokes) that cannot be properly modeled through sequential token processing.

The input sequence then passes through multiple shared Transformer blocks, outputting the shifted right predicted feature sequence for each time step $\{\hat{\mathbf{Z}}_t\}_{t=2}^{T+1}$:

$$\{\hat{\mathbf{Z}}_t\}_{t=2}^{T+1} = \text{TransformerBlocks}(\{\mathbf{Z}_t\}_{t=1}^{T}, \mathbf{M}). \quad (7)$$

Due to numerical differences between dynamic systems, we use nRMSE to calculate the loss function $L_{\text{sim}}$ for a batch during training:

$$\begin{aligned} L_{\text{sim}}^u &= \frac{1}{|B|} \sqrt{\Sigma_{(x,t) \in B} \left( \frac{u^{\text{pred}}(x, t) - u(x, t)}{\sigma_u} \right)^2}, \\ L_{\text{sim}} &= \frac{1}{C} \Sigma_{j \in C} L_{\text{sim}}^j. \end{aligned} \quad (8)$$

This design can effectively capture the temporal evolution of physical fields, achieving high-precision dynamic system predictions and ensuring that systems with different numbers of physical quantities can adapt to this model for temporal regression predictions.

### 3.2. Fine-tuning OmniArch: Enabling Physics-Informed Learning via Equation Supervision

The PDE equations are natural and intuitive 'supervision' methods for real-world physical phenomena. To perform the physical alignment, we incorporate the PDE-Aligner to achieve physics-informed learning. Unlike the pre-training stage, the OmniArch is designed to comply with specific physical laws during fine-tuning. As illustrated in Figure 3 (left), the PDE-Aligner employs a contrastive learning paradigm in the frequency domain.

The key insight is that physical evolution manifests distinctively in frequency space—conservation laws constrain energy distribution across modes, while different PDEs exhibit characteristic spectral signatures. By operating in this domain, PDE-Aligner captures these fundamental patterns more effectively than spatial approaches. It compares the dynamic system's semantics with statistical characters of the frequency domain, where the dynamical system descriptions, namely equations, boundaries, initial conditions, and other physical priors, are encoded into a representation $E_{\text{text}}(\mathcal{P})$.

To characterize physical evolution, we acquire the initial state $u(x, t_0)$ and the current state $u(x, t_i)$ of the physical field, applying the Fourier Encoder to obtain their $k$-th frequency domain representations $\hat{u}_K(k, t_i)$ and $\hat{u}_K(k, t_0)$. The phase difference $\Delta\phi = (\hat{u}_K(k, t_i) \cdot \hat{u}_K^*(k, t_0))/(|\hat{u}_K(k, t_i)||\hat{u}_K(k, t_0)|)$ captures wave propagation and dispersion characteristics, while the magnitude ratio $R = |\hat{u}_K(k, t_i)|/|\hat{u}_K(k, t_0)|$ quantifies energy transfer across scales—both serving as physics-aware fingerprints of the underlying PDE. Thus, we have the alignment loss function as:

$$
\begin{aligned}
L_{\text{Align}} &= L_{\text{eq}} + \lambda L_{\text{E}}, \\
L_{\text{eq}} &= \mathcal{S}(\, E_{\text{text}}(\mathcal{P}), \boldsymbol{\Psi}[\Delta\phi, R]^\top), \\
L_{\text{E}} &= |\sum_K R - 1|.
\end{aligned} \tag{9}
$$

where $\lambda$ is a hyperparameter balancing the energy conservation loss. The energy term $L_{\text{E}}$ enforces Parseval's theorem, ensuring physical consistency in the frequency domain. By minimizing the alignment loss function $L_{\text{Align}}$, the PDE-Aligner aligns the changes in the physical field with the textual descriptions within the constraints of energy conservation.

In the fine-tuning stage (Figure 3 Right), the pre-trained

PDE-Aligner serves as a physics-aware discriminator, helping OmniArch distinguish between different physical regimes encountered during pre-training. The fine-tuning loss $L_{\text{ft}} = L_{\text{sim}} - L_{\text{eq}}$ encourages predictions that are both accurate (via $L_{\text{sim}}$) and physically consistent with the specified PDE system (via $L_{\text{eq}}$), effectively steering the model toward the correct physical behavior among many learned dynamics.

## 4. Experiments

### 4.1. Dataset and Baselines

**Dataset.** We collect 1D, 2D, and 3D datasets from the public PDEBench and PDEArena. The 1D datasets include: (1) **CFD**, generated by the compressible Navier-Stokes equation with velocity ($V_x$), density, and pressure. (2) **Bur.**, the Burgers' equation with velocity. (3) **Diff.**, the diffusion-sorption equation with concentration ($\rho$). (4) **Adv.**, the advection equation with velocity ($V_x$). (5) **Reac.** the reaction-diffusion equation with concentration ($\rho$). The 2D datasets include: (6) **CFD**, generated by the compressible Navier-Stokes equation with velocities ($V_x, V_y$), density, and pressure. (7) **Reac.**, the reaction-diffusion equation with activator ($u$) and inhibitor ($v$). (8) **SWE**, the shallow-water equation with velocities ($h$). (9) **Incom.**, generated by 2D Inhomogeneous, Incompressible Navier-Stokes equations, with velocities ($V_x, V_y$) and particles. The 3D datasets include: (10) **CFD**, generated by the compressible Navier-Stokes equation with velocities ($V_x, V_y, V_z$), density, and pressure. (11) **Maxw.**, the Maxwell equation with electric displacement ($D_x, D_y, D_z$) and magnetic field ($H_x, H_y, H_z$). More details can be found in Appendix C.

**Baselines.** The baselines are divided into two categories: (1) *Task-specific expert models*, which include Physics-Informed Neural Networks (PINNs) (Raissi et al., 2019), U-Net (Ronneberger et al., 2015), and Fourier Neural Operator (FNO) (Li et al., 2020), all of which require training from scratch for each specific case (each equation/coefficient, etc.). (2) *Unified pre-training models*, which include PDEformer-1 (Ye et al., 2024), Multiple Physics Pre-training (MPP) (McCabe et al., 2023), SWIN-transformer (Liu et al., 2021) used for the ORCA task, the large size pretrained checkpoint of Poseidon (Herde et al., 2024) and DPOT (Hao et al., 2024). More details on the baselines are provided in Appendix D.

**Training Details.** The OmniArch model uses single-layer encoders and decoders for data of various dimensions, with the LLaMA model (trained from scratch) as the shared Transformer architecture. The PDE-Aligner employs the pre-trained Fourier encoder from OmniArch to encode physical fields and the pre-trained BERT model to encode PDE captions. Additional training details are in Appendix E.

Table 1: The nRMSE on various PDEs. We evaluate base-size(-B) and large-size(-L). The previous state-of-the-art performance is underlined and our best performance is **bolded**.

| Methods | 1D | | | | | 2D | | | | 3D | |
|---|---|---|---|---|---|---|---|---|---|---|---|
| | CFD | Adv. | Bur. | Diff. | Reac. | CFD | Reac. | SWE | Incom | CFD | Maxw. |
| *Baselines -Task specific Expert Models* | | | | | | | | | | | |
| **PINNs** | / | 0.8130 | 0.9450 | 0.2200 | 0.2140 | / | 1.6000 | 0.0170 | / | / | / |
| **U-Net** | 2.6700 | 0.7760 | 0.3201 | 0.1507 | 0.0026 | 1.0700 | 0.8401 | 0.0830 | 1.1200 | 0.7989 | 0.2999 |
| **FNO** | 1.4100 | 0.0091 | 0.0174 | 0.0017 | 0.0005 | 0.2060 | 0.1203 | 0.0044 | 0.2574 | 0.3052 | 0.1906 |
| *Baselines - Unified Pre-training and Fine-tuning* | | | | | | | | | | | |
| **PDEformer-1** | – | 0.0043 | 0.0095 | – | 0.0009 | – | – | – | – | – | – |
| **ORCA-SWIN-B** | – | – | – | – | – | / | 0.8201 | 0.0062 | / | – | – |
| **MPP-AVIT-B** | – | – | – | – | – | 0.0227 | 0.0106 | 0.0024 | / | – | – |
| **MPP-AVIT-L** | – | – | – | – | – | 0.0178 | 0.0098 | 0.0022 | / | – | – |
| **Poseidon-L** | – | – | – | – | – | 0.1079 | 0.0949 | 0.0243 | – | – | – |
| **DPOT-L** | – | – | – | – | – | 0.0112 | 0.0263 | 0.0451 | – | 0.4321 | – |
| *Full Pre-Training on 1D,2D,3D Data* | | | | | | | | | | | |
| **OmniArch-B(Ours)** | 0.0340 | 0.0238 | 0.0089 | 0.0020 | 0.0006 | 0.0196 | 0.0158 | 0.0016 | 0.1726 | 0.5209 | 0.2834 |
| **OmniArch-L(Ours)** | 0.0250 | 0.0182 | 0.0063 | 0.0015 | 0.0004 | 0.0148 | 0.0105 | 0.0014 | 0.1494 | 0.4531 | 0.2268 |
| *+ PDE-Aligner Fine-tuning* | | | | | | | | | | | |
| **OmniArch-B(Ours)** | 0.0302 | 0.0201 | 0.0071 | 0.0017 | 0.0003 | 0.0153 | 0.0102 | 0.0015 | 0.0955 | 0.4032 | 0.1813 |
| **OmniArch-L(Ours)** | **0.0200** | **0.0041** | **0.0032** | **0.0006** | **0.0002** | 0.0125 | **0.0084** | **0.0012** | **0.0827** | 0.3723 | **0.1671** |
| **std. $\pm$** | 0.0031 | 0.0012 | 0.0004 | 0.0001 | 0.0001 | 0.0017 | 0.0004 | 0.0003 | 0.0023 | 0.0443 | 0.0197 |
| **Improvement ↑** | **98.70%** | 4.65% | 66.32% | 64.75% | 60.00% | – | 14.28% | 45.45% | 67.87% | – | 12.32% |

**Notes**: Symbol '/' means model did not converge while '–' means model not applicable to this dataset.

## 4.2. Results and Analysis

OmniArch is designed to support multi-scale, multi-physics, and flexible physics alignment. Table 1 presents the normalized root mean square error (nRMSE) across various PDEs for different methods.

**Multi-Physics Results.** (1) Compared with Task-specific Expert Models. PINNs, U-Net, and FNO require training from scratch for each specific equation or coefficient. While FNO shows strong performance, PINNs and U-Net struggle with convergence and accuracy in some cases (Like the CFD-1D, and CFD-2D). (2) Compared with Unified Pre-training Models. PDEformer-1 exhibits proficiency in specific 1D equations but fails to generalize beyond its formulation structure. MPP and ORCA-SWIN leverage 2D pre-training and fine-tuning, improving generalization, yet their effectiveness remains constrained by the diversity of the pre-training data. Poseidon enables single-step inference at arbitrary timesteps, though its accuracy still leaves room for improvement. DPOT successfully transfers knowledge from 2D to 3D CFD through weight sharing, but it lacks support for 1D CFD and its performance on non-CFD physics systems requires further enhancement. (3) Om-

niArch Performance. OmniArch, pre-trained on 1D, 2D, and 3D data, demonstrates superior performance across all evaluated datasets. Both the base (B) and large (L) versions of OmniArch outperform existing models, validating its robustness in multi-physics contexts. To validate our architectural design choices, we conduct ablation studies on the Temporal Mask mechanism (Table 2). The results confirm that our Temporal Mask, which enables full attention among physical quantities within each timestep, significantly outperforms standard causal masking across various multi-physics systems. (4) PDE-Aligner Fine-tuning. Fine-tuning with PDE-Aligner significantly enhances OmniArch's accuracy, particularly for complex datasets. This step utilizes a pre-trained Fourier encoder and BERT-base-cased model, ensuring precise alignment between physical fields and PDE descriptions. Table 3 quantifies the impact of PDE-Aligner across different dimensions, showing consistent improvements of over 20% compared to pre-training alone. OmniArch demonstrates substantial performance gains over baselines, with up to 98.70% improvement on CFD-1D and notable enhancements across other PDEs.

**Ablation Study on Masking Strategies.** As illustrated in Table 2, the superiority of Temporal Mask (18-20% improve-

Table 2: Ablation study on masking strategies

| Dataset | Causal Mask | No Mask | Temporal Mask |
|---------|-------------|---------|---------------|
| 2D Incom. | 0.0277 | 0.0285 | **0.0227** |
| 2D CFD | 0.0198 | 0.0205 | **0.0148** |
| 3D CFD | 0.1842 | 0.1923 | **0.1494** |

| Methods | Shock | KH | OTVortex |
|---------|-------|-----|----------|
| **FNO** | 0.7484 | 1.0891 | 0.5946 |
| **U-Net** | 1.6667 | 0.1677 | 0.4217 |
| **MPP-L** | 0.3243 | 1.3261 | 0.3025 |
| **OmniArch-L** | **0.2126** | **0.2763** | **0.1718** |

Table 4: The Performance on Zero-shot PDEs.

ment) reveals a fundamental insight: multi-physics systems require simultaneous rather than sequential processing of coupled variables. This advantage is most pronounced in 3D CFD, where the complex interplay between five physical quantities (velocities, density, pressure) demands holistic attention patterns.

Table 3: Impact of PDE-Aligner on model performance (OmniArch-L)

| Configuration | 1D PDEs | 2D PDEs | 3D PDEs |
|---------------|---------|---------|---------|
| Pre-training only | 0.0103 | 0.0440 | 0.3399 |
| Fine-tuning w/o Aligner | 0.0073 | 0.0345 | 0.3432 |
| Fine-tuning w/ Aligner | **0.0056** | **0.0262** | **0.2697** |
| **Improvement** | 23.3% | 24.1% | 21.4% |

**Impact of PDE-Aligner.** We report the impact of PDE-Aligner in Table 3, where the consistent 22% improvement across dimensions suggests that PDE-Aligner serves as more than a physics constraint—it helps OmniArch disambiguate between different physical regimes learned during pre-training. Notably, the similar improvement ratios across 1D-3D indicate that physical alignment is dimension-agnostic, validating our unified architecture design.

**Multi-scale Results.** In Figure 4, we present the multi-scale inference performance of OmniArch-Base and OmniArch-Large on the 2D Incom. Dataset. Due to the frequency truncation capability of the Fourier Encoder, OmniArch can handle inputs of varying grid sizes without requiring re-training. In the red-shaded area, the nRMSE decreases as the grid size becomes smaller. Conversely, in the blue-shaded area, the nRMSE slightly increases. However, even with a grid size of 512, the maximum nRMSE remains below 0.2. In the rollout settings, a grid size of 256 sometimes leads to better or comparable performance to a grid size of 128. The non-monotonic relationship between grid resolution and error (red vs. blue regions in Figure 4) reveals an intriguing property of frequency-domain learning: OmniArch naturally identifies the intrinsic resolution of physical phenomena. The optimal performance at intermediate resolutions (128-256) suggests the model has learned to distinguish between meaningful physical scales and numerical artifacts. Additional visualizations are provided in Appendix H.5.

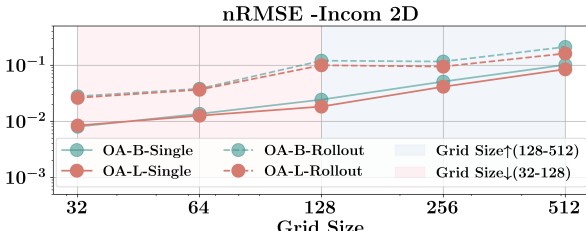

Figure 4: The multi-scale capability.

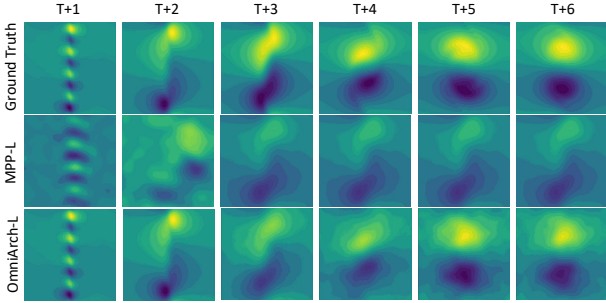

Figure 5: Zero-shot prediction results (Rollout) of OmniArch-L and MPP-L on KH dataset. Displaying time steps T+1 to T+6, the top row shows ground truth data, while the middle and the bottom row illustrate MPP-L's and OmniArch-L's predictions respectively.

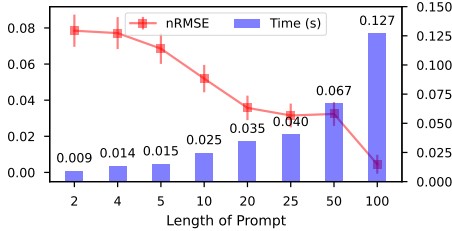

Figure 6: In-context learning on SWE with OmniArch-B.

Table 5: nRMSE for Inverse Problems.

| Methods | Forcing | Buoyancy |
|---------|---------|----------|
| **MPP** | $0.2 \pm 0.008$ | $0.78 \pm 0.006$ |
| **OmniArch** | $0.16 \pm 0.005$ | $0.73 \pm 0.012$ |
| **Scratch** | $0.39 \pm 0.012$ | $0.83 \pm 0.027$ |

**Flexible Physics Alignment**. To verify the PDE-Aligner's ability to perceive physical information, we equipped it with a classification head to classify physical fields. In Figure 7, the PDE-Aligner can perceive physical field categories based on equation text information and physical field features, and the classification accuracy rate exceeds 0.94 on all ten categories. More details are in Appendix F.3.

**Zero-shot Performance.** Our examination of 2D PDE predictions, as illustrated in Figure 5, reveals that OmniArch effectively captures both low- and high-frequency patterns even in zero-shot scenarios, surpassing former 2D models like MPP. MPP often misses key features, leading to erroneous representations of the primary physics and failed rollouts. Details of zero-shot dataset are in Appendix C.2.

As shown in Table 4, nRMSE scores indicate that all models, except OmniArch, tend to underperform in zero-shot transfer. This suggests that OmniArch's use of Fourier Encoders and unified training approach enhances its ability to generalize across different PDEs. By leveraging flexible grid inputs and dynamic observation windows during pre-training, OmniArch effectively captures the underlying physics of the observed field states, which may not be adequately addressed by methods adhering strictly to explicit grid and temporal dependencies. The 4-7× error reduction compared to MPP in zero-shot scenarios (Table 4) indicates that OmniArch has learned transferable physical operators rather than dataset-specific patterns. The success on shock-dominated flows (Shock, KH)—notoriously difficult for neural methods—demonstrates that frequency-domain representations capture discontinuities more effectively than spatial approaches.

**In-Context Learning.** After autoregressively pre-trained on various dynamic systems, we observe that OmniArch could learn neural operators within the observations of several time steps, which is similar to the in-context learning in Large Language Models. Here, we define the given time-series of observations as *PDE Prompt*. Our approach varies the prompt length from 2 tokens (derived from a 50 time step interval) to 100 tokens (from a 1 time step interval). More details are in Appendix H.2.

**Fine-tuning for Inverse Problems.** Demonstrating a model's capability to infer hidden physical parameters from known equations is a critical test of its ability to learn underlying physics. The results in Table 5 demonstrate that

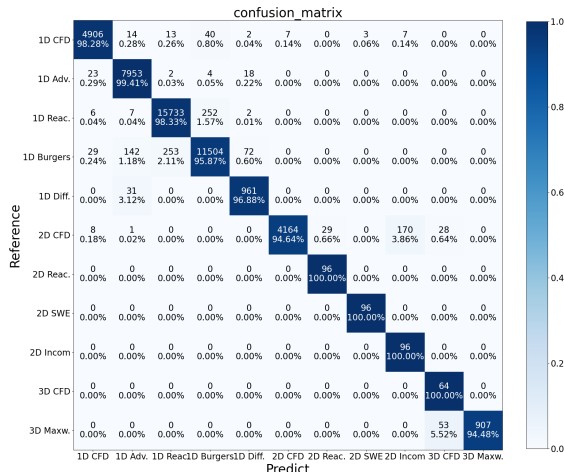

Figure 7: The confusion matrix of the PDE-Aligner classification results.

OmniArch outperforms MPP in parameter estimation tasks, with lower RMSE values indicating more accurate predictions. Models trained from scratch yield the highest errors, underscoring the effectiveness of our fine-tuning approach. This evidence supports the notion that OmniArch is not only proficient in forward simulations but also exhibits superior performance in deducing hidden dynamics within complex systems. More details are in Appendix H.3.

**Other Results.** In addition to the primary experiments, we include more rollout case studies in Appendix H.4 and report the inference-time GPU Memory usage compared with baselines in Appendix H.7. We also include ablation studies for training settings in Appendix G and the detailed performance for CFD PDEs in Appendix H.6. These additional evaluations highlight OmniArch's robustness and accuracy in complex physical simulations, surpassing other state-of-the-art models.

## 5. Conclusion

In this study, we introduced a pioneering foundation model for scientific computing, specifically tailored for the resolution of partial differential equations (PDEs). By integrating this model with a novel PDE-Aligner for fine-tuning, we have established new state-of-the-art benchmarks across a comprehensive suite of tasks within the PDEBench. Additionally, we investigated the zero-shot learning capabilities of our pre-trained model, uncovering a degree of transferability that mirrors the emergent properties found in large-scale language models. Despite the successes, we recognize the challenges posed by 3D PDE systems to our OmniArch model, which may leave for future research. We envisage that OmniArch will serve as a cornerstone for developing foundation models in the domain of PDE learning, fostering a significant convergence between scientific machine learning (SciML) and broader deep learning disciplines.

## Acknowledgement

This work was supported by the National Science and Technology Major Project(No.2022ZD0117800), and Young Elite Scientists Sponsorship Program by CAST(No.2023QNRC001). This work was also sponsored by CAAI-Huawei MindSpore Open Fund (CAAIXSJLJJ2023MindSpore12) and developed on openl community. Thanks for the computing infrastructure provided by Beijing Advanced Innovation Center for Big Data and Brain Computing.

## Impact Statement

OmniArch represents a significant advancement in scientific computing. It unifies multi-scale and multi-physics PDE-solving capabilities within a single foundation model framework. This unified approach has profound implications for accelerating scientific discovery and engineering applications across domains such as fluid dynamics, weather forecasting, and materials science. The model's demonstrated ability to handle diverse physical systems and grid resolutions while maintaining physical consistency could dramatically reduce the computational resources required for complex simulations in industrial and research settings. Additionally, there are many potential societal consequences of our work, none of which we feel must be specifically highlighted here.

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

## Supplementary Material for:

OMNIARCH: Building Foundation Model For Scientific Computing

## Contents

Table 6: Table of notations

| | Basic Notations |
|---|---|
| $x, t$ | Spatial and Temporal coordinate (time) |
| $\Delta t$ | Time interval |
| $T$ | Total time steps |
| $D$ | Total dimensions of physical fields |
| $C$ | Number of physical fields |
| $B$ | Batch size |
| $\mathcal{U}$ | Physical field inputs |
| $u(x^{(d)}, t)$ | Physical field of dimension $d$ at spatial coordinate $x$ and time $t$ |
| $\boldsymbol{\Psi}(\cdot)$ | Linear projection |
| | **OmniArch Related Notations** |
| $\mathcal{F}, \mathcal{F}^{-1}$ | Fourier transformation and its inverse |
| $F$ | Components (modes) in the frequency domain |
| $\hat{\mathcal{U}}$ | Physical field frequency domain representation |
| $k$ | Frequency Variable |
| $K$ | Number of retained Fourier modes (cut-off frequency) |
| $\hat{u}(k, t)$ | Fourier transform of $u(x, t)$ at frequency $k$ and time $t$ |
| $\hat{u}_K(k, t)$ | Truncated Fourier modes (TopK modes) at frequency $k$ and time $t$ |
| $\hat{u}_K^{\text{pred}}(k, t)$ | Predicted Fourier modes at frequency $k$ and time $t$ |
| $u^{\text{pred}}(x, t)$ | Predicted physical field at spatial coordinate $x$ and time $t$ |
| $f(\cdot)$ | Integral operator |
| $\alpha_{i,t}$ | Attention weights at spatial coordinate $i$ and time $t$ |
| $\mathcal{R}(\cdot)$ | Real-valued embedding function of the frequency domain features |
| $U_t, V_t$ | Physical field embedding token from $\mathcal{R}$ at time $t$ |
| $\mathbf{Z}_t$ | Input sequence consist of grouped embeddings from $U_t, V_t$ |
| $\hat{\mathbf{Z}}_t$ | Shifted right predicted feature sequence |
| $\mathbf{M}$ | Temporal mask used in Transformer Blocks |
| $\sigma_u$ | Normalization factor $\|u\|_2^2 + \epsilon$ for nRMSE calculation |
| $L_{\text{sim}}^u$ | Normalized RMSE loss: $\sqrt{\mathbb{E}[(u^{\text{pred}} - u)^2]}/\sigma_u$ |
| $L_{\text{sim}}$ | Mean nRMSE across all physical fields |
| | **PDE-Aligner Related Notations** |
| $\mathcal{P}$ | PDE text description (captions) |
| $E_{\text{text}}(\cdot)$ | Text encoder used for PDE captions |
| $\Delta\phi$ | Phase difference between the initial state and current state |
| $R$ | Amplitude ratio between two states |
| $\lambda$ | The hyperparameter balancing the the energy conservation loss |
| $L_{\text{eq}}$ | Similarity between text embedding and physical embedding |
| $L_{\text{E}}$ | Energy conservation loss |
| $L_{\text{Align}}$ | PDE-Aligner training loss |
| $L_{\text{ft}}$ | OmniArch fine-tune loss |

## A. Table of notations

A table of notations is given in Table 6.

## B. Limitations

Despite its advancements, OmniArch remains fundamentally data-driven, and its interpretability requires further improvement, even with the PDE-Aligner enhancing physical prior alignment. Constraints in computational power and data

availability have limited OmniArch's scalability, affecting its generalization capabilities, particularly in complex and abrupt dynamical systems such as 3D tasks and shock wave PDEs. Addressing these limitations is crucial for further development and broader applicability in scientific and engineering contexts.

## C. Dataset details

### C.1. OmniArch Pre-training Dataset

**Pre-training Stage**. We structured the PDEBench data into distinct training, validation, and testing subsets. For one-dimensional (1D) PDEs, the training dataset comprises a selection from the CFD-1D, ReacDiff, Advection, Burgers, and diff-sorp datasets. From these, we reserve a random 10% sample of trajectories as the in-domain test set for each respective PDE equation. The Shock Tube Equation is designated as the out-of-domain test set. Additionally, the test portions of the reacdiff and diff-sorp datasets are utilized as part of the test set.

In the two-dimensional (2D) PDE case, we allocate 90% of trajectories from the CFD, diff-react, NSincom, and shallow water datasets for training. The remaining 10% form the in-domain test set. The Shock Tube, Kelvin-Helmholtz instability (KH), and Tolman-Oppenheimer-Volkoff (TOV) scenarios are included as out-of-domain test sets.

For three-dimensional (3D) PDEs, 90% of trajectories from the CFD-3D dataset are utilized for training, with the remaining 10% serving as the in-domain test set. The complete datasets for blastwave and turbulence simulations are used as out-of-domain test sets. The Details of our pre-training dataset can be found in Table 7.

Table 7: Data Statistics for OmniArch Pre-training

| | Dataset | #Train | #Validation | #Physical quantities | $N_t$ | $N_s$ |
|---|---|---|---|---|---|---|
| **1D** | **CFD** | 45000 | 5000 | velocities $V_x$, density, pressure | 100 | 1024 |
| | **Reac.** | 144000 | 16000 | concentration $\rho$ | 200 | 1024 |
| | **Adv.** | 72000 | 8000 | velocities $V_x$ | 200 | 1024 |
| | **Bur.** | 108000 | 12000 | velocities $V_x$ | 200 | 1024 |
| | **Diff.** | 9000 | 1000 | concentration $\rho$ | 100 | 1024 |
| **2D** | **CFD** | 39600 | 4400 | velocities $V_x, V_y$, density, pressure | 21 | 512 |
| | **Reac.** | 900 | 100 | activator $u$, inhibitor $v$ | 100 | 128 |
| | **Incom** | 900 | 100 | velocities $V_x, V_y$, particle | 1000 | 256 |
| | **SWE** | 900 | 100 | velocities $h$ | 100 | 128 |
| **3D** | **CFD** | 630 | 70 | velocities $V_x, V_y, V_z$, density, pressure | 21 | 128 |
| | **Maxw.** | 8640 | 960 | electric displacement $D_x, D_y, D_z$ magnetic field $H_x, H_y, H_z$ | 8 | 64 |

### C.2. Dataset For Zero-shot Learning

We choose three test datasets from PDEBench to validate the zero-shot ability of our model. They all belong to two-dimensional compressible Navier-Stokes equations but are different fluid phenomena that exhibit distinct physical mechanisms and characteristics. Brief introductions and details of the datasets are as follows:

- **OTVortex:** The Orszag-Tang Vortex system is a compressible flow problem that generates highly complex vortex structures through the careful selection of initial conditions. The dataset includes one example, which is a $1024 \times 1024$ resolution physical field evolved over 101 time steps with a time interval of 0.01.

- **2D Shock:** Shock waves are characterized by abrupt changes in flow properties resulting from sudden discontinuities

in fluid flow, such as rapid changes in pressure, temperature, and density. The dataset includes one example, which is also a $1024 \times 1024$ resolution physical field evolved over 101 time steps with a time interval of 0.01.

- **2D KH:** The Kelvin-Helmholtz instability is a fluid instability that occurs at the interface between two fluid layers with different velocities or densities. This dataset consists of seven examples generated based on different parameters $M, dk$, and $Re$. Each is a $1024 \times 1024$ resolution physical field evolved over 51 time steps with a time interval of 0.1. We conducted experiments on all samples and averaged the results.

## D. Baseline implementation details

In our experiments, we adopt the benchmarking framework provided by PDEBench (Takamoto et al., 2022) and select three well-established methods for comparative analysis. Furthermore, we have incorporated the Multiple Physics Pre-training (MPP) model into our comparative analysis to address the need for retraining that is inherent to the aforementioned methods when faced with novel sets of conditions, the detailed training hyperparameters of FNO, U-Net, and PINN is provided in Table 8, following PDEbench (Takamoto et al., 2022). The first hyperparameter of U-Net is the unroll steps (denoted as **us**), and the second is the train steps (denoted as **ts**). The hyperparameters shared by both FNO and U-Net are the initial steps (denoted as **is**) and batch size (denoted as **bs**). The hyperparameter in PINNs is the hidden size (denoted as **hid**). The learning rate, shared by FNO, U-Net, and PINNs, is denoted as **lr**.

Table 8: Setting details when training FNO, U-Net, and PINN, * means shared setting for FNO and U-Net, shared setting for FNO, U-Net, and PINN is denoted with a symbol †.

|  |  | FNO | | U-Net | | is* | bs* | PINNs | lr† |
|  |  | modes | width | us | ts |  |  | hid |  |
|---|---|---|---|---|---|---|---|---|---|
| 1D | Adv. | 12 | 20 | 20 | 200 | 10 | 50 | 40 | 0.001 |
|  | Bur. | 12 | 20 | 20 | 200 | 10 | 50 | 40 | 0.001 |
|  | CFD | 12 | 20 | 20 | 100 | 10 | 50 | 40 | 0.001 |
|  | Diff. | 12 | 20 | 20 | 101 | 10 | 50 | 40 | 0.001 |
|  | Reac. | 12 | 20 | 20 | 101 | 10 | 50 | 40 | 0.001 |
| 2D | CFD | 12 | 20 | 20 | 21 | 10 | 20 | 40 | 0.001 |
|  | Reac. | 12 | 20 | 20 | 101 | 10 | 5 | 40 | 0.001 |
|  | SWE | 12 | 20 | 20 | 101 | 10 | 5 | 40 | 0.001 |
|  | Incom | 12 | 20 | 20 | 101 | 10 | 20 | 40 | 0.001 |
| 3D | CFD | 12 | 20 | 20 | 21 | 10 | 5 | 40 | 0.001 |
|  | Maxw. | 12 | 20 | 7 | 8 | 7 | 5 | 40 | 0.001 |

**Physics-Informed Neural Networks (PINNs)** (Raissi et al., 2019). PINNs utilize neural networks to solve differential equations by embedding physical laws into a multi-objective optimization framework, minimizing PDE residuals and boundary/initial condition errors (Cuomo et al., 2022).

**U-Net** (Ronneberger et al., 2015). U-Net, designed for biomedical image segmentation, uses an encoder-decoder structure for context capture and precise localization (Siddique et al., 2021; Du et al., 2020). We adapt U-Net into 1D and 3D forms to analyze spatio-temporal patterns in physical fields.

**Fourier Neural Operator (FNO)** (Li et al., 2020). FNO pioneers in learning function-to-solution mappings by parameterizing integral kernels in the Fourier domain, enabling efficient and accurate resolution-invariant neural operators.

**PDEformer-1** (Ye et al., 2024). PDEformer-1 is a neural solver capable of simultaneously addressing various types of 1D partial differential equations. It uses a graph Transformer and implicit neural representation (INR) to generate mesh-free predicted solutions.

**Multiple Physics Pre-training (MPP)** (McCabe et al., 2023). MPP extends PDEBench's 2D physics scenarios to learn versatile features for predicting dynamics across various physical systems and comprises pre-training and fine-tuning phases, warranting its inclusion in our comparative analysis.

**ORCA-SWIN** (Shen et al., 2023; Liu et al., 2021). ORCA fine-tunes the SWIN Transformer for different PDEs by first aligning the embedded feature distribution of the target PDE data with the pre-training modality, and then refining the model on this aligned data to effectively leverage shared knowledge across various PDEs.

## E. OmniArch implementation details

### E.1. Pre-training OmniArch

In our training process, the following strategies or decisions were made:

- **Pre/Post Norm**: Pre-norm

- **Norm Type**: RMS Norm Type

- **Architecture**: Decoder-Only

- **Attention-Type**: Multi-scaled Attention

- **Position Embedding**: RoPE

- **Casual Masking**: True- We only evaluate the loss on the T + 1 physical fileds prediction.

- **Hidden Size**: 1024

- **initializer_range**: 0.02

- **intermediate_size**: 4096

- **num_attention_heads**: 16

Table 9: Detailed setting of hyperparameters in pre-training the base and large models. The batch sizes, modes, and widths are provided as lists, with values corresponding to 1D, 2D, and 3D data respectively.

| Hyperparameters | Base | Large |
|---|---|---|
| #Layers | 12 | 24 |
| Hidden Size | 768 | 1024 |
| #Heads | 12 | 16 |
| Intermediate Size | 3072 | 4096 |
| Batch Sizes | [42,3,1] | [32,2,1] |
| Modes | [12,12,12] | [12,12,12] |
| Widths | [8,8,8] | [8,8,8] |
| Learning Rate | 0.0001 | 0.0001 |
| Scheduling Method | Cosine Annealing | Cosine Annealing |

We trained two different sizes of model: base and large, which primarily differ in the number of layers, hidden sizes, number of heads, and intermediate sizes, as detailed in Figure 9. For the base model, we selected batch sizes of [42, 3, 1] for the 1D, 2D, and 3D trajectories, respectively. These batch sizes represent the maximum capacities our acceleration devices could handle while maintaining the ratio of data trajectories. This configuration allows for optimal training efficiency by minimizing idle time and maximizing device utilization. For the large model, due to its significantly increased size, we adjusted the batch sizes to [32, 2, 1] to ensure that the GPU memory is fully utilized. This reduction in batch sizes accommodates the larger model's memory requirements while still enabling effective training across the different dimensions of data trajectories.

### E.2. Fine-tuning OmniArch

Fine-tuning is performed on an A40 GPU cluster, which has 40GiB of memory per device. The fine-tuning settings for each dataset are shown in Table 10. We set the learning rate to 1e-5, which results in fast convergence. Using 2 GPUs in Distributed Data-Parallel mode, we fine-tune each dataset for a maximum of 30 epochs and apply early stopping.

Table 10: Detailed Fine-tuning Settings: The table provides the learning rate, width, modes, and batch size for 1D, 2D, and 3D data.

| Dims | learning rate | width | modes | batch size | Scheduling Method |
|------|---------------|-------|-------|------------|-------------------|
| **1D** | 1e-5 | 8 | 12 | 64 | Cosine Annealing |
| **2D** | 1e-5 | 8 | 12 | 8 | Cosine Annealing |
| **3D** | 1e-5 | 8 | 12 | 2 | Cosine Annealing |

### E.3. Parameter Efficiency Analysis

Table 11: Static Parameter Distribution (Millions)

| Model Component | OmniArch-B (316M) | OmniArch-L (672M) |
|-----------------|-------------------|-------------------|
| Shared Backbone | 138 (43.7%) | 435 (64.7%) |
| 1D Encoder/Decoder | 0.3 | 0.4 |
| 2D Encoder/Decoder | 7.0 | 9.0 |
| 3D Encoder/Decoder | 171 | 227 |

Table 12: Active Parameters During Task Execution (Millions)

| Model | PDE Type | | |
|-------|----------|---|---|
| | 1D PDEs | 2D PDEs | 3D PDEs |
| OmniArch-B | 138 | 144 | 308 |
| OmniArch-L | 435 | 445 | 663 |

As illustrated in Table 11 and Table 12, the parameter distribution reveals OmniArch's hierarchical design philosophy. Three key observations emerge: (1) The shared backbone dominates the parameter count (43.7–64.7%), facilitating cross-dimensional knowledge transfer while requiring only modest modality-specific additions (0.3–227M). (2) For 2D tasks (MPP's primary domain), OmniArch-B activates merely 144M parameters—a 24.1% increase over MPP-B's 116M that brings three key advantages: (a) unified architecture reduces system complexity, (b) enables latent cross-modal learning, and (c) provides future-proof extensibility. (3) The scaling pattern shows intelligent allocation—3D processing requires 2.1–2.7× more dedicated parameters than 2D, reflecting its inherent higher dimensionality while maintaining efficient reuse of the shared backbone.

## F. PDE-Aligner implementation details

### F.1. PDE-Aligner Pre-training Dataset

**PDE-Aligner equation augmentation**. Given the significant imbalance between equation caption data and physical field data, a single equation can yield a multitude of physical field simulations. To augment equation captions effectively, it is crucial to preserve the equation's solutions and boundaries while adhering to physical laws and exploring a wide array of possible substitutions. To achieve this, we have developed a five-step augmentation pipeline: *Equation Rewriting*, *Form Transformation*, *Linear Combination*, *Symbol Substitution*, and *Physical Checking*:

- **Equation Rewriting.** We apply mathematical identities to modify the equation, ensuring the core properties remain intact.

- **Form Transformation.** We transform equations between differential and integral forms and employ techniques such as Green's functions to broaden the equation's representations.

- **Linear Combination.** For systems of equations, we derive new variants through linear combinations, enriching the dataset without altering the system's nature.

- **Symbol Substitution.** We systematically swap variables with alternative symbols, such as replacing $x$ with $\xi$, to maintain consistency and avoid ambiguity.

- **Physical Checking.** A panel of GPT-4-based experts evaluates the augmented equations, filtering out those that do not align with physical principles.

Leveraging the first four steps, we generate 200 augmented instances per equation type. Subsequently, during the Physical Checking phase, we select the top 50% of these examples based on quality for pre-training. Representative samples of the augmented examples are available in Appendix F.2.

Additionally, we randomly sample the numerical distributions of different physical quantities at two distinct time steps within the physical field to represent the field's temporal variations. Each set of two-step physical field data is paired with a corresponding enhanced equation text to form a single data instance. This approach is used to compile a comprehensive pre-training dataset for the PDE-Aligner.

### F.2. Examples of Generated PDEs

#### F.2.1. BURGERS 1D

- Original form:

$$\partial_t u(t, x) + \partial_x(u^2(t, x)/2) = \nu/\pi \partial_{xx} u(t, x), \quad x \in (0, 1), t \in (0, 2],$$

$$u(0, x) = u_0(x), \ x \in (0, 1),$$

- After augmented:

$$0.77 \int \left( \frac{\partial}{\partial t} v(t, x) + \frac{\partial}{\partial x} \frac{v^2(t, x)}{2} \right) dt = \frac{0.77\nu \int \frac{\partial^2}{\partial x^2} v(t, x)\, dt}{\pi}$$

$$0.73 t v(0, x) = 0.73 t v_0(x)$$

- Explanation: We replace $u$ with $v$ and $\partial_t$ with $\frac{\partial}{\partial t}$. We integrate and multiply some factors on both sides of the equation at the same time.

#### F.2.2. ADVECTION

- Original form:

$$\partial_t u(t, x) + \beta \partial_x u(t, x) = 0, \quad x \in (0, 1), t \in (0, 2],$$

$$u(0, x) = u_0(x), \ x \in (0, 1),$$

- After augmented:

$$1.45 \int \left( c\frac{\partial}{\partial x} A(t, x) + \frac{\partial}{\partial t} A(t, x) \right) dt = 0$$

$$A(0, x) = A_0(x)$$

- Explanation: We replace $u$ with $A$, $\partial_t, \partial_x$ with $\frac{\partial}{\partial t}, \frac{\partial}{\partial x}$, and $\beta$ with $c$. We integrate and multiply some factors on both sides of the equation at the same time.

**F.2.3. CFD-1D**

- Original form:

$$\partial_t \rho + \nabla \cdot (\rho \mathbf{v}) = 0,$$

$$\rho(\partial_t \mathbf{v} + \mathbf{v} \cdot \nabla \mathbf{v}) = -\nabla p + \eta \triangle \mathbf{v} + (\zeta + \eta/3)\nabla(\nabla \cdot \mathbf{v}),$$

$$\partial_t \left[ \epsilon + \frac{\rho v^2}{2} \right] + \nabla \cdot \left[ \left( \epsilon + p + \frac{\rho v^2}{2} \right) \mathbf{v} - \mathbf{v} \cdot \sigma' \right] = 0,$$

- After augmented:

$$\varrho(t,x)\frac{\partial}{\partial x}\mathbf{w}(t,x) + \frac{\partial}{\partial t}\varrho(t,x) = 0 \tag{10}$$

$$0.61\left(\mathbf{w}(t,x)\frac{\partial}{\partial x}\mathbf{w}(t,x) + \frac{\partial}{\partial t}\mathbf{w}(t,x)\right)\varrho(t,x) =$$

$$0.61\eta\frac{\partial^2}{\partial x^2}\mathbf{w}(t,x) + 0.61\left(\chi + \frac{\eta}{3}\right)\frac{\partial^2}{\partial x^2}\mathbf{w}(t,x) - 0.61\frac{\partial}{\partial x}p(t,x) \tag{11}$$

- Explanation: We replaced many symbols, such as replacing $\nabla$ with $\partial_t$ and $\triangle$ with $\frac{\partial^2}{\partial x^2}$,. We integrate and multiply some factors on both sides of the equation at the same time. We also swapped the order of some items, such as $\zeta + \eta/3$.

Table 13: Detailed Data Information: The total amounts of training data, sampled training data, total validation data, and sampled validation data are presented as lists. These lists correspond to 1D, 2D, and 3D data respectively.

| Dims | Total training | Sampled training | Total validation | Sampled Validation |
|------|----------------|------------------|------------------|--------------------|
| 1D | 218T | 378K | 269M | 42K |
| 2D | 3.13T | 42K | 38M | 5K |
| 3D | 748K | 0.63K | 9K | 0.07K |

**F.3. Pre-training process of PDE-Aligner**

In our architecture, the PDE-Aligner is divided into two components: a text encoder and a physics encoder. The text encoder utilizes the pre-trained albert-math model (Reusch et al., 2022), which is highly capable of processing LaTeX-encoded PDE captions due to its extensive training on a large corpus of LaTeX data. For the physics encoder, we employ the pre-trained Fourier encoder from OmniArch, known for its strong ability to capture physical field features. We adopt a large-batch contrastive learning approach similar to SimCLR (Chen et al., 2020). The training involves a stochastic sampling strategy with an equal probability (50%) of selecting either canonical PDE captions sourced directly from textbooks or augmented PDE captions. The latter is assumed to enhance the text encoder's generalization capabilities while retaining critical PDE information in textual form. The weights of the text encoder and physics encoder are fixed during the PDE-Aligner training process. The training data details for PDE-Aligner are shown in Table 13, and the hyperparameter settings are provided in Table 14.

During the fine-tuning phase, the PDE-Aligner evaluates the alignment of gold-standard PDE captions with the state of physical fields at each step of generator G's decoding process. The resulting rewards are averaged over the temporal dimension and finalized upon the completion of inference. The intuition behind the PDE-Aligner fine-tuning is to help OmniArch distinguish the patterns behind different PDE systems. To verify the PDE-Aligner's ability to perceive physical information, we equipped it with a classification head to classify physical fields. The results, shown in Figure 8, indicate that the PDE-Aligner effectively aligns with physical laws.

Table 14: Detailed Hyper-parameters Settings: The init learning rate, optimizer, scheduler, hidden size, trainable params, total params, steps, and GPU hrs are presented as lists.

| Hyper-parameters | Value |
|---|---|
| Init Learning Rate | 1e-4 |
| Optimizer | Adam |
| Scheduler | Cosine Annealing |
| Hidden Size | 768 |
| Trainable Params | 1.2M |
| Total Params | 195M |
| Steps | 37k |
| GPU hrs | 75 |

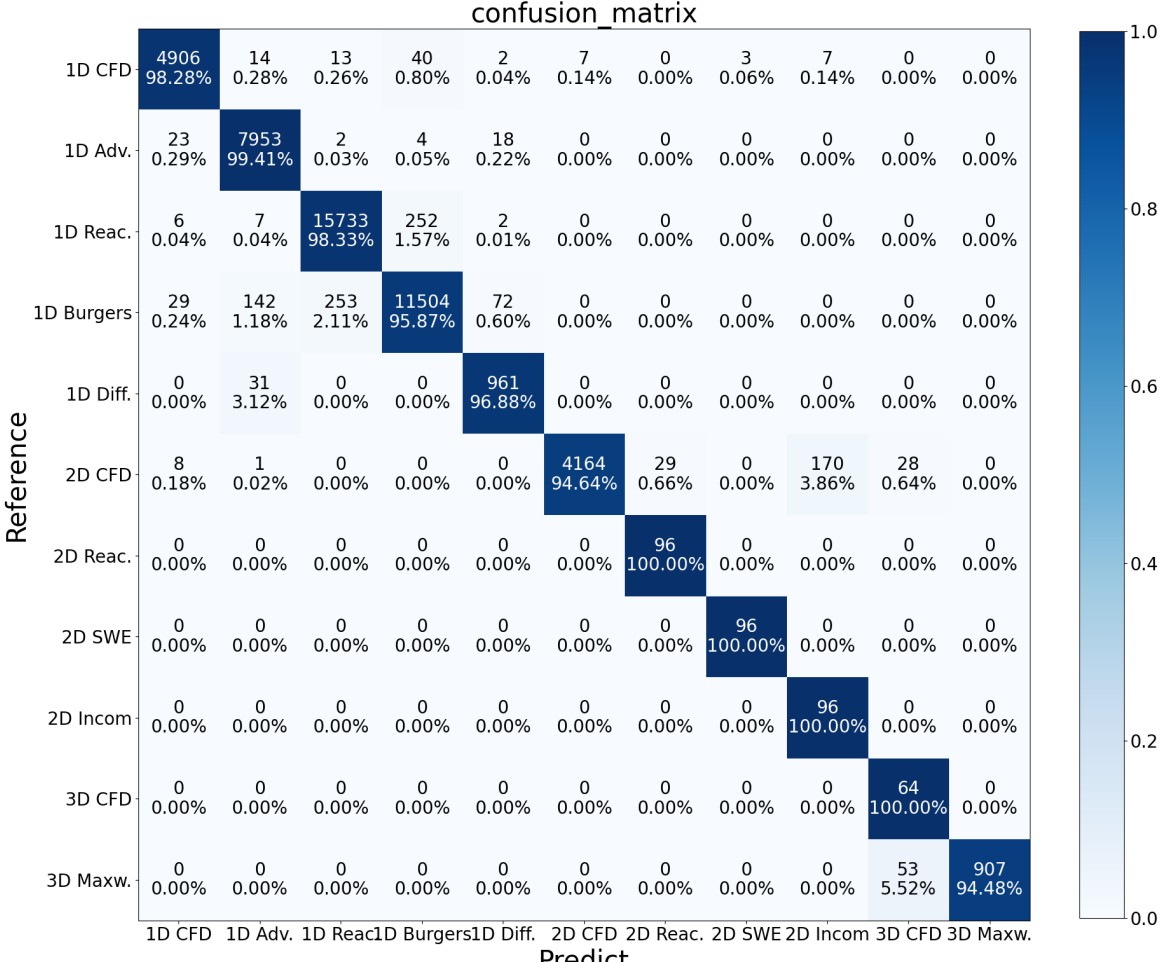

Figure 8: **The confusion matrix of the PDE-Aligner classification results.** PDE-Aligner can perceive physical field categories based on equation text information and physical field features, and the classification accuracy rate exceeds 0.94 on all ten categories.

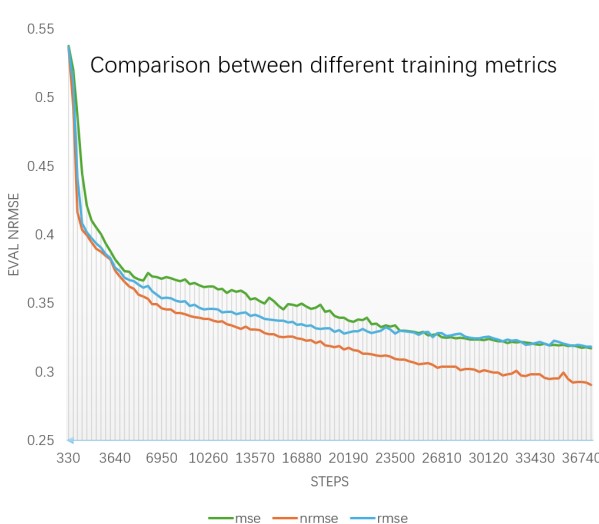

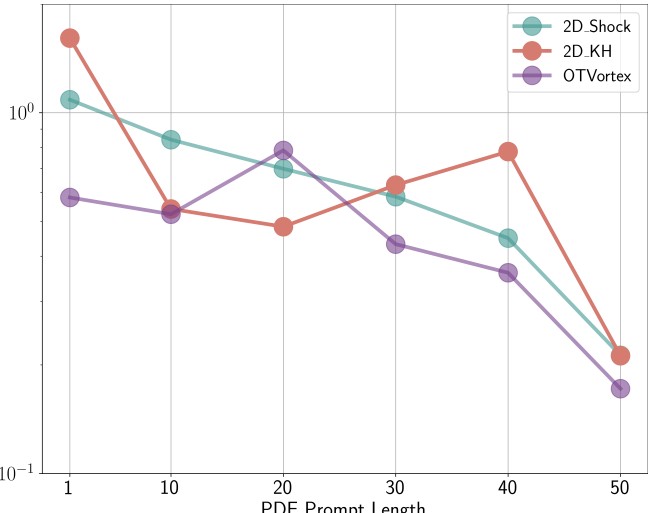

Figure 9: The training loss curve using different metrics.

Figure 10: Zero-shot learning nRMSE for T+1 timesteps with varying context lengths.

## G. Further ablation study

We have conducted an ablation study on batch-wise nRMSE. We use nRMSE, RMSE, and MSE respectively as loss functions in the training process. We found that nRMSE leads to a more unified loss scale for different PDEs, benefiting OmniArch's convergence. Table 15 shows nRMSE yielded lower training losses compared to MSE and RMSE (up to 9.3% improvement). While we did encounter gradient calculation issues in extreme cases, these were mitigated by adding a small $\epsilon$ to the squared norm of the true labels (averaged over the spatial dimension) . Utilizing nRMSE as a training loss function aims to simultaneously reduce all channels, irrespective of their relative numerical values. The loss curve is shown in Figure 9.

Table 15: Training loss metrics ablation study.

| Steps | MSE | RMSE | nRMSE |
|---|---|---|---|
| 10K | 0.3624 | 0.3458 | **0.3386** (-2.08%) |
| 20K | 0.3371 | 0.3289 | **0.3175** (-3.47%) |
| 30K | 0.3240 | 0.3225 | **0.3005** (-6.82%) |
| 40K | 0.3181 | 0.3183 | **0.2887** (-9.83%) |

## H. More results

### H.1. Zero-shot Learning Capability

Our examination of 2D PDE predictions reveals that, in contrast to task-tuned models, the OmniArch model adeptly captures both low- and high-frequency patterns in in-domain PDEs such as Reaction Diffusion, CFD, Shallow Water, and Incompressible NS. Task-tuned models often miss key features, occasionally leading to erroneous representations of the primary physics. For out-of-domain PDEs, delineated by a red-dotted box in the figure, we evaluated the models' ability to predict unseen PDEs without fine-tuning or parameter adjustment. While task-tuned models consistently failed at this zero-shot learning task, OmniArch successfully predicted essential low-frequency background patterns, though it struggled with high-frequency details. Details on the zero-shot dataset, including shock wave, Kelvin-Helmholtz (KH), and Orszag-Tang Vortex (OTVortex) phenomena, are provided in Appendix C.2.

In our zero-shot learning evaluation, we explore the minimum number of time steps necessary to formulate accurate neural operators. We also probe the OmniArch model's ability to generalize to new physics scenarios without parameter adjustments. As indicated in Table 4 and Figure 10, a longer temporal context typically enhances model performance, resulting in lower nRMSE scores across tasks. Notably, our model exhibits impressive zero-shot learning capabilities, maintaining robustness against mesh and temporal interpolation variations, even with fewer than 20 time steps of context.

### H.2. Dynamic Prompt Length for Efficient Inference

We examine the trade-off between inference speed and accuracy using dynamic prompt lengths in our model. The goal is to determine whether shorter prompts can accelerate inference times on the CPU without significantly sacrificing precision.

Our approach varies the prompt length from 2 tokens (derived from a 50 time step interval) to 100 tokens (from a 1 time step interval) to predict physical fields at $u_{101}$. As shown in Figure 6, longer prompts yield higher precision with less variance, while shorter prompts can expedite inference by up to 10 times compared to full-length prompts. In particular, our model demonstrates an inherent ability to learn temporal differences from the input sequence, negating the need for explicit time-step inputs.

### H.3. Fine-tuned for Inverse Problems

Demonstrating a model's capability to infer hidden physical parameters from known equations is a critical test of its ability to learn underlying physics. Following the methodology of MPP (McCabe et al., 2023), we evaluate our model on two inverse problems for incompressible Navier-Stokes equations: 1) Forcing Identification, and 2) Buoyancy Determination.

Table 16: RMSE for Parameter Estimation in Inverse Problems.

| Methods | Forcing | Buoyancy |
|---|---|---|
| **MPP** | $0.2 \pm 0.008$ | $0.78 \pm 0.006$ |
| **OmniArch** | $0.16 \pm 0.005$ | $0.73 \pm 0.012$ |
| **Scratch** | $0.39 \pm 0.012$ | $0.83 \pm 0.027$ |

The results in Table 16 demonstrate that OmniArch outperforms MPP in parameter estimation tasks, with lower RMSE values indicating more accurate predictions. Models trained from scratch yield the highest errors, underscoring the effectiveness of our fine-tuning approach. This evidence supports the notion that OmniArch is not only proficient in forward simulations but also exhibits superior performance in deducing hidden dynamics within complex systems.

### H.4. Rollout Predictions

We perform rollout experiments to compare the performance of the Fourier Neural Operator (FNO) model and our proposed OmniArch model, as depicted in Figure 11, 12, 13, 14. Our findings indicate that OmniArch demonstrates superior adherence to the underlying physics laws in the initial timesteps, as opposed to merely replicating patterns from other trajectories. This improved fidelity is likely a result of fine-tuning with PDE-Aligner, which isolates the model from the influences of alternate PDE systems, thereby enhancing the model's ability to generalize physical dynamics.

### H.5. Multi-scale Inference Results

To thoroughly evaluate the multi-scale forecasting capabilities of OmniArch, extensive experiments were conducted across four different grid resolutions: $32 \times 32$, $64 \times 64$, $128 \times 128$, and $256 \times 256$. Figure 15 presents the visualization results at $T + 50$ time step on the Incom dataset. These results demonstrate OmniArch's robust ability to accurately capture local patterns across varying grid sizes, confirming its effectiveness in handling multi-scale data without losing detail or accuracy.

### H.6. More results in different problem settings

We tested our model on CFD-2D problems under various settings of the Navier-Stokes equations to evaluate its performance across different scenarios. The goal was to determine the robustness and adaptability of our model, OmniArch, compared to

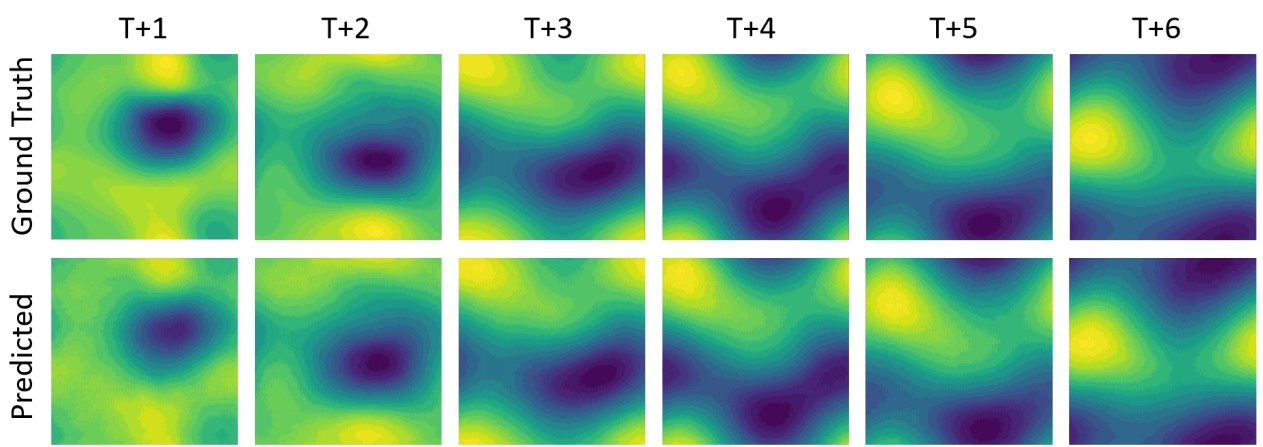

Figure 11: Prediction results of OmniArch on CFD-2D dataset. Displaying time steps T+1 to T+6, the top row shows ground truth data, and the bottom row illustrates OmniArch's predictions.

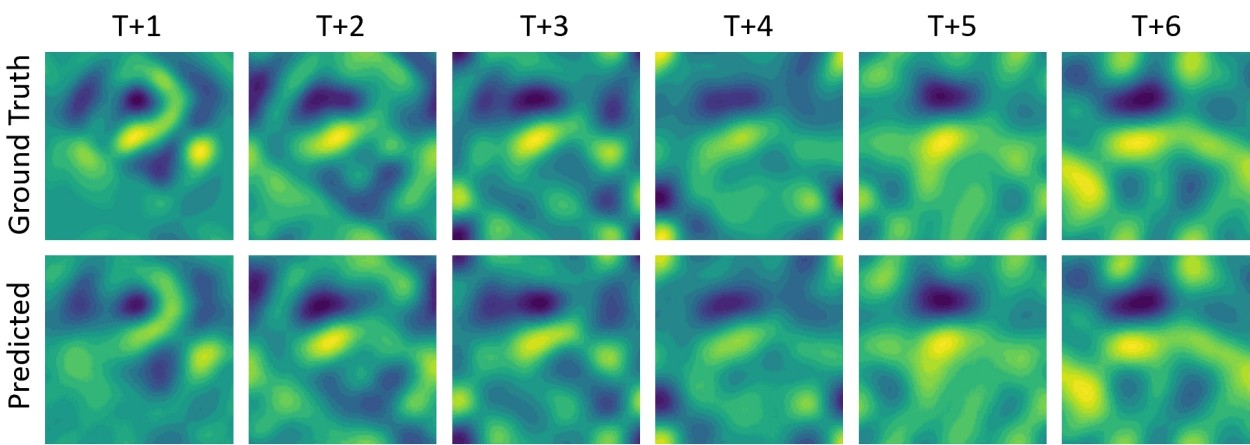

Figure 12: Prediction results of OmniArch on CFD-2D dataset. Displaying time steps T+1 to T+6, the top row shows ground truth data, and the bottom row illustrates OmniArch's predictions.

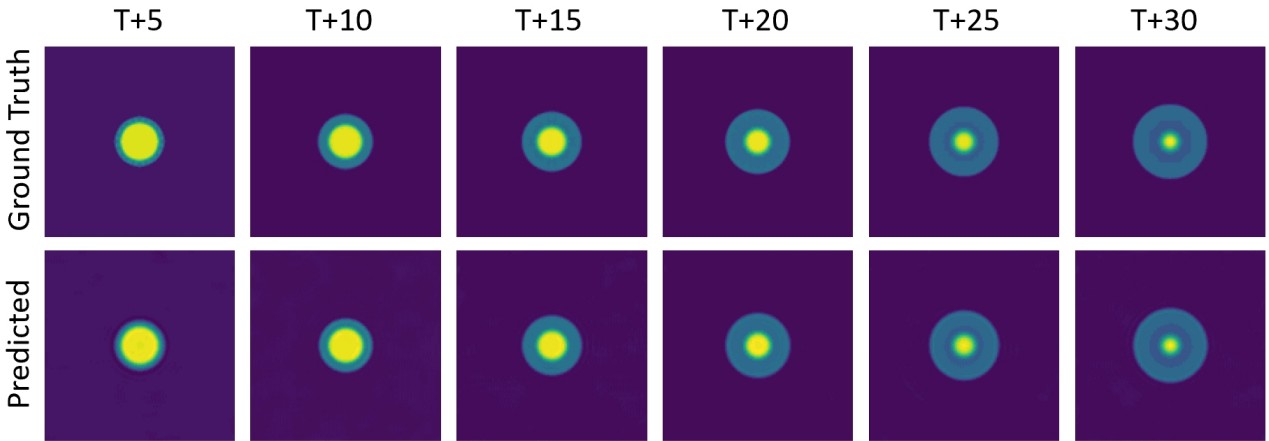

Figure 13: Prediction results of OmniArch on SWE dataset. Displaying time steps T+5 to T+30, the top row shows ground truth data, and the bottom row illustrates OmniArch's predictions.

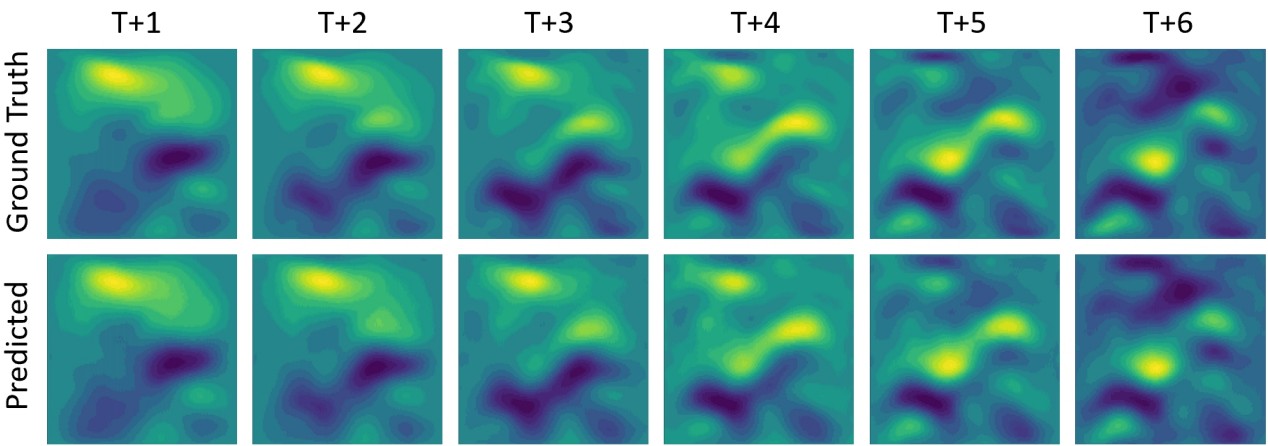

Figure 14: Prediction results of OmniArch on Incom dataset. Displaying time steps T+1 to T+6, the top row shows ground truth data, and the bottom row illustrates OmniArch's predictions.

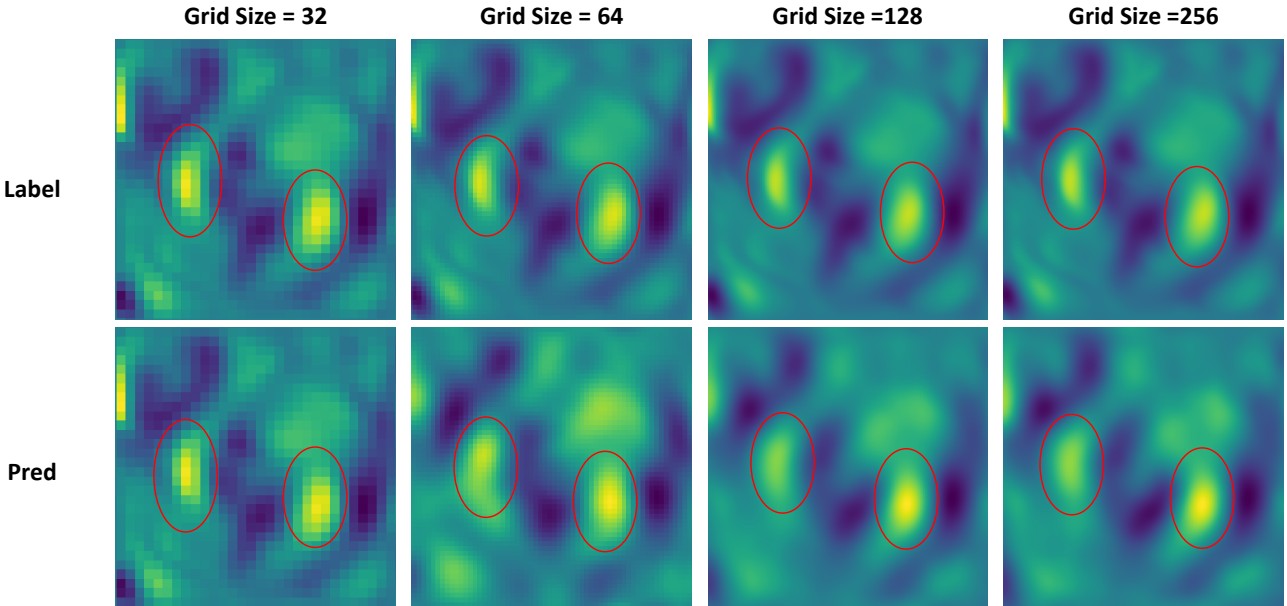

Figure 15: Multi-scale results of OmniArch-Large with different grid sizes.

other state-of-the-art models like MPP, FNO, and U-Net.

Table 17 summarizes the performance results of our model, OmniArch (FT), against MPP (FT), FNO, and U-Net across multiple problem settings. These settings include variations in Mach number ($M$), viscosity ($\eta$), and diffusivity ($\xi$), both in inviscid and turbulent conditions with random periodic boundary conditions.

Table 17: The different problem settings in 2D Navier Stokes equation performance.

| Problem Settings | OmniArch(FT) | MPP(FT) | FNO | U-Net |
|---|---|---|---|---|
| $M = 0.1$, inviscid Rand periodic | **0.1600** | 0.5866 | 0.38 | 0.66 |
| $M = 0.1$, $\eta = \xi = 0.01$ Rand periodic | **0.1215** | 0.5286 | 0.17 | 0.71 |
| $M = 0.1$, $\eta = \xi = 0.1$ Rand periodic | **0.0273** | 0.5761 | 0.36 | 5.1 |
| $M = 1.0$, $\eta = \xi = 0.01$ Rand periodic | **0.1301** | 0.5096 | 0.196 | 0.36 |
| $M = 1.0$, inviscid Rand periodic | **0.1387** | 0.5391 | 0.35 | 0.47 |
| $M = 1.0$, $\eta = \xi = 0.1$ Rand periodic | **0.0308** | 0.5033 | 0.098 | 0.92 |
| $M = 0.1$, inviscid Turb periodic | **0.2219** | 0.3949 | 0.16 | 0.19 |
| $M = 1.0$, inviscid Turb periodic | **0.1624** | 0.5412 | 0.43 | 0.14 |

These results consistently show that OmniArch performs better across various settings, demonstrating its robustness and effectiveness. The performance advantage of OmniArch is evident across different Mach numbers, viscosity, and diffusivity settings, both in inviscid and turbulent conditions. These findings highlight the model's capability to generalize and maintain high accuracy in diverse and challenging CFD scenarios.

### H.7. GPU Memory Usage and Inference Time

We also report the runtime and memory usage in Table 18. OmniArch consistently uses less GPU memory than MPP across all model sizes, demonstrating its efficiency in resource utilization. While FNO and U-Net have lower GPU memory usage and faster inference times, OmniArch's performance remains competitive, particularly considering its ability to handle a wider range of PDE tasks across 1D, 2D, and 3D domains.

Table 18: The runtime and memory usage between different models.

| Model | Size | GPU Memory | Inference Time |
|---|---|---|---|
| OmniArch | Tiny | 671MB | 0.0125s |
| | Small | 866MB | 0.0129s |
| | Base | 1591MB | 0.0136s |
| | Large | 3109MB | 0.0248s |
| MPP | Tiny | 1378MB | 0.0387s |
| | Small | 1532MB | 0.0390s |
| | Base | 1620MB | 0.0391s |
| | Large | 3270MB | 0.0831s |
| FNO | - | 690MB | 0.0018s |
| U-Net | - | 830MB | 0.0027s |

### H.8. Comparison with Traditional Solvers

Our benchmarks reveal three key advantages of OmniArch over traditional solvers:

- **Resolution Invariance**: While FDM computation time scales quadratically ($O(n^2)$) with grid resolution, OmniArch maintains nearly constant inference time (23-26ms) due to its fixed-frequency processing in the spectral domain. This yields exponential speedup ($155\times$ at $512\times512$) for high-resolution simulations.

Table 19: Computational Efficiency Comparison (2D Advection)

| Resolution | FDM Time/Step (ms) | OmniArch Time (ms) | Speedup | Relative Error |
|---|---|---|---|---|
| 64×64 | 1.123 | 23.567 | 0.048× | 1.24× |
| 128×128 | 15.264 | 23.820 | 0.641× | 1.18× |
| 192×192 | 75.360 | 24.098 | 3.128× | 1.15× |
| 256×256 | 254.027 | 24.083 | 10.55× | 1.12× |
| 320×320 | 583.218 | 23.866 | 24.44× | 1.09× |
| 384×384 | 1130.561 | 23.453 | 48.20× | 1.07× |
| 448×448 | 2272.206 | 23.677 | 95.96× | 1.05× |
| 512×512 | 4073.472 | 26.212 | 155.4× | 1.03× |

- **Accuracy Preservation**: Despite dramatic speed improvements, OmniArch maintains comparable accuracy with relative error consistently below 1.25× of FDM results. The error margin decreases at higher resolutions (1.03× at 512×512), suggesting better performance in practical high-fidelity scenarios.

- **Generalization Capability**: Unlike traditional methods requiring re-discretization for new PDEs, OmniArch's unified architecture achieves this performance across multiple physics domains (Navier-Stokes, Advection-Diffusion, etc.) without algorithmic modifications, as demonstrated in Section 4.2.

The results validate our design choice of spectral-domain processing - while sacrificing some interpretability inherent to mesh-based methods, OmniArch gains orders-of-magnitude efficiency improvements crucial for large-scale multi-physics simulations. This trade-off aligns with emerging trends in scientific ML where learned simulators complement (rather than replace) traditional methods for specific high-throughput applications.

## I. More Discussions

### I.1. Meta-Learning vs. Scaling Laws in PDE Solving

While meta-learning methods (Chen et al., 2022; Huang et al., 2022; Cho et al., 2023) address generalization through gradient-based adaptation, OmniArch explores an orthogonal axis: scaling laws for in-context learning. The distinction mirrors "learning to optimize" versus "learning from data" paradigms—meta-PINNs refine their optimization trajectory for new PDEs, whereas foundation models leverage scale to discover physics-aware primitives. These approaches need not compete; future work might hybridize them. We may imagine meta-learning the hypernetworks of a foundation model.

