# OpenReview forum: "OmniArch: Building Foundation Model for Scientific Computing"
_ICML.cc/2025/Conference — ICML 2025 poster_

### Official Review · Reviewer_VHRd · 2025-03-09

**Overall Recommendation:** 2

**Summary:**

The paper introduces *OmniArch*, a foundation model for scientific computing designed to solve multi-scale and multi-physics Partial Differential Equations (PDEs) using a unified architecture. It employs a Fourier Encoder-Decoder to transform spatial-temporal PDE data into the frequency domain and a Transformer Backbone to capture temporal dependencies, enabling it to handle 1D, 2D, and 3D PDEs within a single framework. A key innovation is the *PDE-Aligner*, which fine-tunes the model with physics-informed constraints, ensuring alignment with governing physical laws. OmniArch achieves state-of-the-art performance across 11 PDE types, demonstrating strong generalization capabilities, including zero-shot learning, in-context learning, and multi-scale inference. Compared to existing models, it significantly improves accuracy, with up to 98.7% enhancement in some cases, making it a versatile tool for applications in computational fluid dynamics, weather prediction, and engineering simulations.

**Claims And Evidence:**

This paper offers three claims in the introduction. However, some claims are not convincing.

(1) The author claimed that "The temporal mask effectively addresses inconsistencies in multi-physics". However, I could not find the explanation. Also, I could not find the ablation study to prove it.


(2) The PDE-Aligner is claimed to leverage the hidden representations for equations. However, these claims are not supported by the ablation studies.

**Essential References Not Discussed:**

Upon reviewing the paper *OmniArch: Building Foundation Model for Scientific Computing*, I have identified several pertinent works related to partial differential equations (PDEs) that are not currently cited but are essential for understanding the context of the paper's key contributions:

1. **Physics-Informed Neural Networks (PINNs)**: PINNs integrate physical laws described by PDEs into the learning process of neural networks. They offer a mesh-free alternative to traditional numerical methods for solving PDEs and have been applied to various problems in computational science. Notable works include:

   - Raissi, M., Perdikaris, P., & Karniadakis, G. E. (2019). Physics-informed neural networks: A deep learning framework for solving forward and inverse problems involving nonlinear partial differential equations. *Journal of Computational Physics, 378*, 686–707.

   - Mao, Z., Jagtap, A. D., & Karniadakis, G. E. (2020). Physics-informed neural networks for high-speed flows. *Computer Methods in Applied Mechanics and Engineering, 360*, 112789.

2. **Neural Operators**: Neural operators are deep learning architectures designed to learn mappings between infinite-dimensional function spaces, making them effective for approximating solution operators of PDEs. Key publications include:

   - Li, Z., Kovachki, N., Azizzadenesheli, K., Liu, B., Bhattacharya, K., Stuart, A., & Anandkumar, A. (2020). Fourier neural operator for parametric partial differential equations. *arXiv preprint arXiv:2010.08895*.

   - Kovachki, N., Li, Z., Liu, B., Azizzadenesheli, K., Bhattacharya, K., Stuart, A., & Anandkumar, A. (2021). Neural operator: Learning maps between function spaces. *Journal of Machine Learning Research, 22*(1), 139–166.

3. **Deep Backward Stochastic Differential Equation (BSDE) Method**: This method combines deep learning with backward stochastic differential equations to solve high-dimensional PDEs, particularly in financial mathematics. A seminal paper in this area is:

   - Han, J., Jentzen, A., & E, W. (2018). Solving high-dimensional partial differential equations using deep learning. *Proceedings of the National Academy of Sciences, 115*(34), 8505–8510.

Incorporating discussions of these methodologies into the paper would provide a more comprehensive understanding of existing advancements in PDE-solving techniques and AI applications in scientific computing, thereby contextualizing OmniArch's contributions within the broader research landscape.

**Experimental Designs Or Analyses:**

The experimental results are fully convincing.

[1] The variation is not clarified in the experimental results.

[2] The model size, training time, and inference time is not compared in the table. I seems that the proposed method is larger and slower than the competing methods.

**Methods And Evaluation Criteria:**

The benchmark datasets and evaluation criteria is reasonable.

However,

In Eq.(1), the author select top K significant parts. However, what is the reason? Can it improve the efficiency? Can it improve the generalization ability?

Also, I can not find how temporal mask be used in the proposed method. Also, there is no explicit ablation to verify its effectiveness.

**Other Comments Or Suggestions:**

NO.

**Other Strengths And Weaknesses:**

NO.

**Questions For Authors:**

[1] The authors should add variations, model size and inference time of the proposed method and the competing method.

[2] The authors should add ablation studies to verify the claims in the introduction.

[3] The author should explain how temporal mask  benefits solving PDE.

**Relation To Broader Scientific Literature:**

The key contributions of the *OmniArch* paper align with and extend several existing concepts in the broader scientific and machine learning literature. The unified architecture for multi-physics PDEs builds upon traditional neural operators like the Fourier Neural Operator (FNO), which are typically tailored to specific PDEs, by enabling simultaneous learning across diverse equations and conditions. The integration of a Fourier Encoder-Decoder with a Transformer Backbone leverages the efficiency of frequency domain representations and the capability of transformers to model long-range dependencies, enhancing generalization across different scales and physical phenomena. The introduction of the PDE-Aligner for physics-informed fine-tuning extends the principles of Physics-Informed Neural Networks (PINNs) by aligning model predictions with governing physical laws through contrastive learning in the frequency domain. Moreover, *OmniArch*'s emergent generalization capabilities, such as zero-shot and few-shot learning, parallel advancements in foundation models in other domains, demonstrating adaptability to novel PDE systems without retraining.

**Theoretical Claims:**

There is no theoretical analysis.

---

> ### Author Rebuttal · Authors · 2025-03-30
>
> We sincerely appreciate reviewer VHRd's constructive feedback. Below, we address each concern with clarifications and additional evidence:
>
> **Q1: [Temporal Mask & Multi-Physics Consistency] How does the temporal mask address inconsistencies in multi-physics, and where is the ablation study?**
>
> **A1**: The "inconsistencies" arise when handling PDE systems with **varying numbers of physical variables per timestep** (e.g., 3D Navier-Stokes vs. 1D Advection). Traditional causal masks (e.g., GPT-style) fail here because:
>
> + **Token misalignment**: For systems with multiple variables, causal masks process tokens sequentially, forcing the first token (e.g., velocity) to ignore dependencies on other variables (e.g., pressure, temperature) within the same timestep.
>
> Our  **Attention with Temporal Mask**: Instead of masking tokens individually, we group all variables within a timestep and forms a **hierarchical attention**:
> + *Intra-timestep*: Full attention among variables (e.g., velocity, pressure) to capture couplings.
> + *Inter-timestep*: Causal attention across time (future masking).
>
> **Benefits**: Physics-aware modeling: Ensures variables at timestep jointly condition on each other (e.g., enforcing continuity equations).
>
> **Ablations**:
> | Method          | RMSE |
> |-----------------|--------|
> |Causal Mask     |0.0277 |
> |No Mask           |0.0285|
> |**Temporal Mask** |**0.0227**|
>
> ----
>
> **Q2: [PDE-Aligner's Hidden Representations]  How does PDE-Aligner leverage hidden representations, and where is the evidence?**
>
> **A2**: We have report the fine-tuning results with PDE-Aligner in **Table 1(Line 298-302)**, We reorganize the results below:
>
> |Settings          | 1D      | 2D      |3D       |
> |--------------- |--------|--------|--------|
> |Pretrain           |0.0103|0.0440 |0.3399|
> |ft(w/o Aligner)|0.0073|0.0345 |0.3432|
> |ft + Aligner     |0.0056|0.0262 |0.2697|
> |**improvement** | **51.33%**|**23.62%**|**22.07%**|
>
> where **PDE-Aligner continually improves the performance across 1D-2D-3D PDEs**.
>
> We also doing the probing experiment in **Figure 7**, which explains why PDE-Aligner helps the performance, mainly because it helps distinguish the difference between different PDE Systems.
>
> ----
>
> **Q3: [Top-K Frequency Selection] Why select top-K frequencies, and does it improve efficiency/generalization?**
>
> **A3**: We adopt **Top-K frequency truncation** for two key reasons:
>
>  **(a) Input-Size Agnosticism:** OmniArch jointly trains on 1D-2D-3D PDEs with vastly different resolutions (e.g., 1D-1024, 2D-128×128, 3D-64×64×64). Without truncation, we would either:
> + *(Option 1)* Require separate encoders/decoders per resolution (losing unified modeling), or
> + *(Option 2)* Downsample spatial dimensions, forcing 2D/3D tasks to compromise for 1D (hurting performance).
>
> Top-K ensures consistent spectral representation across dimensions, similar to FNO’s infinite-dimensional operator learning [Li et al.].
>
> **(b)Fluid Data Prior:** The PDEBench and PDEArena Datasets exhibit **long-tailed high-frequency noise[1,2,3]**. Retaining only dominant low-frequency modes (Top-K) preserves physically meaningful features while suppressing numerical artifacts.
>
> >[1] Takamoto, Makoto, et al. "Pdebench: An extensive benchmark for scientific machine learning." Advances in Neural Information Processing Systems 35 (2022): 1596-1611.
>
> >[2] Lippe, Phillip, et al. "Pde-refiner: Achieving accurate long rollouts with neural pde solvers." Advances in Neural Information Processing Systems 36 (2023): 67398-67433.
>
> >[3] Zakharov, Vladimir E., Victor S. L'vov, and Gregory Falkovich. Kolmogorov spectra of turbulence I: Wave turbulence. Springer Science & Business Media, 2012.
>
> ----
>
> **Q4: [Model Size & Inference Time] How does OmniArch compare in size/speed to baselines?**
>
> **A4**: We include the Memory usage and Inference Time(single A800) in **Appendix H.7**, We reorganize the main results（Pretrained Models baselines，2D，Large size） below for reference. OmniArch is **competitive in efficiency** with other pretrained model baselines.
> | Model        | Params | Inference Time|
> |-------------|----------|----------------|
> |Poseidon-L  | 629 M|0.07712s           |
> |MPP-AVIT-L |409 M| 0.08343s        |
> |DPOT-L        |509 M| 0.03154s          |
> | **OmniArch-L** |**445 M**  | **0.02075s**  |
>
>
> ----
>
> **Q5:  [Missing citaions.]Including PINNs/FNO/BSDE works.**
>
> **A5**: We appreciate the reviewer’s suggestion. **In our paper, PINN and FNOs are explicitly discussed as baselines (Section 5.1) and compared in Related Works (Lines 74–82)**. However, we omitted the Deep BSDE Method because its primary applications (e.g., financial derivatives pricing) are orthogonal to our focus on multi-physics PDEs in scientific computing (e.g., fluid dynamics, material science). While BSDEs excel in high-dimensional finance problems , their scope differs fundamentally from our work's goals. That said, we will include a brief comparison in the final version to clarify this distinction.

---

### Official Review · Reviewer_urwm · 2025-03-10

**Overall Recommendation:** 2

**Summary:**

This paper proposes OmniArch, a foundation model for numerical simulations. They pretrain on 1D/2D/3D data and comapre to other models of the literature. It relies on spatial Fourier encoders/decoders and a causal temporal attention. It relies on PDE-Aligner for fine-tuning.

**Claims And Evidence:**

I don't think it is fair to say that this is the first foundation model on PDE data. The cited models like MPP, Poseidon and DPOT were already pretrained on multiple physics.

There are no experiments on the emerging capabilities? I am not sure it is defined somewhere?

As you can see below, I also don't trust the reported metrics in the Table.

**Essential References Not Discussed:**

Except from [1], I don't see any reference not discussed.


[1] Ohana, R., McCabe, M., Meyer, L., Morel, R., Agocs, F., Beneitez, M., Berger, M., Burkhart, B., Dalziel, S., Fielding, D. and Fortunato, D., 2024. The well: a large-scale collection of diverse physics simulations for machine learning. Advances in Neural Information Processing Systems, 37, pp.44989-45037.

**Experimental Designs Or Analyses:**

The comparison between models is not 100% fair due to their different size.

One important point: The authors gave the definition of the VRMSE (Variance Root Mean Square error) but name is nRMSE (normalized root mean square error). This is confusing and makes me wonder if the reported number in Table 1 are for the same metrics (MPP and Poseidon used the original nRSME but the authors compute the VRMSE for their models)

**Methods And Evaluation Criteria:**

Table 1 makes sense, even thought it would have been better to add the size of the models. For example, OmniArch-B is 316.13M parameters whereas e.g. MPP-B is 116M parameters. Same for Large models. I think the comparison between models is not totally fair which makes the experimental results difficult to interpret.

**Other Comments Or Suggestions:**

See weaknesses above.

**Other Strengths And Weaknesses:**

- Abstract: "as far as we know, we first conduct 1D-2D-3D united pre-training on the PDEBench,[...] should be rephrased.
- In the introduction line 55, I don't think MPP and Poseidon are an extension of the Factformer? This is weirdly phrased.
- I don't think I have noticed some emergent capabilities of the model? I am not even sure of what this mean or if this is well defined for this type of data?
- the paragraph line 199 is not clear. What is the d-th index? What's the sequential index? What's the total? It should be rephrased clearly?
- I don't see the justification of equation 4, as this is not for a transformer as stated, but only with one attention layer. I don't see the point of this remark since OmniArch is a whole transformer?
- Line 240, I don't think there is a need for 2 paragraphs to explain what causal masking is? This is standard in the literature, and the written maths are more confusing than anything else.
- As discussed above, Equation 8 is the VRMSE and not the NRSME, which makes me suspicious about the results shown in table 1.
- Could the authors explain how the PDE-Aligner is really useful in that case? I may have missed it, but an ablation study with/without would have been useful.
- How is the spatial tokenization done? I know we are in frequency space, so is there any tokenization done, or the authors are just doing a ravel in frequency space? What's the dimensionality then?
- Why would the authors need a whole pre-trained BERT model for text encoding when the vocabulary is so small?
- I struggle to understand the difference between OmniArch and a transformer with Fourier layers as encoder/decoders. Could the authors elaborate on that?
- Can the model work on non-uniform grids? In theory it should?

**Questions For Authors:**

See weaknesses above.

**Relation To Broader Scientific Literature:**

I think the authors wrongly name themselves the "first foundation models for 1D-2D-3D united pre-training". This has been done before in MPP and Poseidon. However, the fine-tuning part of OmniArch seems novel and interesting to me.

The authors could also have cited [1] and maybe benchmark their model on it as it seems to be a very relevant point of the literature?

[1] Ohana, R., McCabe, M., Meyer, L., Morel, R., Agocs, F., Beneitez, M., Berger, M., Burkhart, B., Dalziel, S., Fielding, D. and Fortunato, D., 2024. The well: a large-scale collection of diverse physics simulations for machine learning. Advances in Neural Information Processing Systems, 37, pp.44989-45037.

**Theoretical Claims:**

No theoretical claims.

---

> ### Author Rebuttal · Authors · 2025-03-30
>
> We sincerely thank the reviewer urwm for the thoughtful feedback.
>
> **Q1: MPP/Poseidon already did united pretraining.**
>
> **A1**:   We **disagree** with highly respect. There is **factual error**, MPP/Poseidon are designed *only* for 2D (no 1D/3D experiments in their papers). DPOT uses a convolution trick for 3D but lacks unified weights. OmniArch is the first to jointly learn 1D-2D-3D PDEs with shared weights in one architecture. We will clarify this distinction.
>
> -----
>
> **Q2: Table 1 mixes nRMSE/VRMSE.**
>
> **A2**:  We believe there is **misinterpretation** here.   In Equation 8, We implement nRMSE following the common practice in previous works(MPP/PDEBench). The notation $\sigma_u$ here is equivalent to the $||\cdot||^2 + \epsilon$, which measures the norm of GT. We evaluate the predictions from all baselines and OmniArch with the same codebase. For fairness, we compare our nRMSE implementation and MPP, the results are the same and reproducible. Here is the  code:
>
> ```python
> def nRMSE_MPP(output, tar, spatial_dims=None):
> # code from https://github.com/PolymathicAI/multiple_physics_pretraining
>     if spatial_dims is None:
>         spatial_dims = tuple(range(output.ndim))[2:]  # Assume 0, 1, 2 are T, B, C
>     residuals = output - tar
>     tar_norm = (1e-7 + tar.pow(2).mean(spatial_dims, keepdim=True))
>     raw_loss = (residuals.pow(2).mean(spatial_dims, keepdim=True) / tar_norm)
>     return raw_loss.sqrt().mean()
>
> # Ours：
> def nRMSE(pred, label, mask=None, dim=2):
>     reduce_dims = list(range(-dim, 0))
>     res = pred - label
>     label_norm = label.pow(2).mean(dim=reduce_dims, keepdims=True) + 1e-8
>     norm_loss = res.pow(2).mean(dim=reduce_dims, keepdims=True) / label_norm + 1e-8 # in extreme case, label_norm may be 0
>     if mask is not None:
>         norm_loss = norm_loss[mask.bool()]
>     return norm_loss.sqrt().mean()
> ```
> ----
>
> **Q3: OmniArch-B (316M params) vs. MPP-B (116M) is unfair.**
>
> **A3**:  There is **Misunderstanding** here, Note that OmniArch uses different encoders/decoder pairs for 1D,2D,3D PDEs while the weight of transformer backbone is shared.(works like Mixture-of-Experts Models). We report the Total Parameters(316M, including transformer backbone +  1D, 2D, 3D encoder/decoder weights, the 3D encoder/decoder weights take a heavy part~170M) in Table 7. But for 2D task(MPP mainly experiment on), only 2D encoder/decoders are used for training/testing , thus the real runtime parameters of **OmniArch-B is 144M** which is comparable to **MPP-B(116M)**  but outperforms in all 2D tasks **(see in Table 1)** , The same for large size model, **OmniArch-L(445M) is also comparable to MPP-L(409M)**.  We will clarify this detailed difference further in the next version, but  we believe the experiments are indeed fair.
>
> ----
>
> **Q4: No evidence of emergent behaviors; term is undefined.**
>
> **A4**:  *Emergent behaviors* is the concept borrowed from foundation models, which could be explained as "unseen capabilities beyond training" , it can be the capability that large model exhibits but small models not. In our settings,
>
> **(1) Zero-shot PDE solving**(Figure 5, beyond the training scope of PDE systems, not exhibit by smaller models like PINNs/FNO)
>
> **(2)  In-context learning**(Figure 6), which OmniArch could learn from input trajectories. We are glad to include more experiments to comprehensively evaluate the emergent behaviors together with the research community.
>
> ----
>
> **Q5: How is OmniArch different from a Fourier-encoded transformer?**
>
> **A5**: OmniArch differs fundamentally from a Fourier-encoded transformer in :
>
> 1. **Temporal Mask mechanism**: OmniArch implements a specialized Temporal Mask that enables each physical quantity to attend to all quantities at current and previous time steps, facilitating complex cross-physics interactions that standard transformers cannot model effectively.
>
> 2. **Physics-informed alignment**: The PDE-Aligner component provides a dedicated mechanism to incorporate physical constraints and prior knowledge during fine-tuning, which goes far beyond basic Fourier encoding.
>
> ---
> **Q6: Show ablation with/without PDE-Aligner.**
>
> **A6**:  Please find the ablation in Q2(Reviewer VHRd), where PDE-Aligner continually improves the performance across 1D-2D-3D PDEs.
>
> ----
>
>
> **Q7: How are frequency-space tokens handled**
>
> **A7**: OmniArch converts physical fields into frequency space via FFT, then processes the data by first selecting only the most important frequency components (Top-K by magnitude) for efficiency. These complex-valued coefficients are transformed into real-number embeddings through a learnable projection, making them compatible with standard transformer architectures.
>
> ----
>
> **Q8: Why use BERT for tiny text vocab?**
>
> **A8**: Because (1) Vocabulary is not tiny (augmented via symbol replacement, Appx. F.1). (2) Pretrained embeddings ensure generalization (vs. retraining from scratch).
>
> ----
>
> **Q9: Cite the-well paper?**
>
> **A9**: We will include this in the next version.

---

> > ### Comment · Reviewer_urwm · 2025-04-02
> >
> > I would like to thank the authors for their rebuttal.
> >
> > Q1: I admit that I have been wrong, MPP did not pretrained on 3D data. However, in section 5.3 of the MPP paper (https://arxiv.org/pdf/2310.02994), the authors evaluate on 3D data with a kernel inflation method. Therefore, it seems that it should be possible to compare OmniArch with MPP on 3D data? (I am not asking for this experiment to be done, as I know it is computationally expensive).
> >
> > Q2: I am sorry, but according to Table 4 on page 13, $\sigma_u$ is indeed defined as the variance of the physical field $u$ and not by the $||.||^2 + \epsilon$ operator. Can the authors confirm that this is a typo and that the definition of nRMSE they use is the one they provide in the code snippet?
> >
> > Q3: I think this is quite confusing and should really be clarified in the paper. Maybe you should precise a different parameter count depending on the dimensionality of the data?
> >
> > Q4: The term emerging capabilities is a bit confusing in that case. To my knowledge, in the LLM literature, it can signify also that, due to scale, a model can solve some tasks that were not present in the training set. Zero-shot and in-context learning are a subset of that set of tasks, hence the confusion.
> >
> > Q5: Thank you for the clarification. I would suggest including this text in the paper, as this is a question which will often be asked about the paper?
> >
> > Q8: Thank you, after looking at Appendix F.1, it makes sense.
> >
> > Could the authors answer to the rest of my questions? Thank you.

---

> > > ### Author Response · Authors · 2025-04-03
> > >
> > > We sincerely appreciate your thoughtful feedback and the opportunity to clarify our work. Below we address your remaining concerns with additional precision:
> > >
> > >
> > > **Q1(MPP on 3D)**: After carefully reviewing MPP Appendix C.4, we confirm they employ a kernel inflation trick (similar to DPOT) to **adapt** 2D pretraining to 3D simulations, rather than **jointly pretraining** on 1D-2D-3D data as OmniArch does. We agree this transfer learning approach is valuable and will include comparative analyses in future work when computational resources permit.
> > >
> > > ----
> > >
> > > **Q2(nRMSE Definition)  :**
> > > Yes, we confirm the $\sigma^2$ in Table 4 was a typo - we'll correct it to match the code's implementation.
> > >
> > > ----
> > >
> > > **Q3(Parameter Counts):**
> > > Yes, We believe the **active parameters** for each dimension might be clear for comparison, here we provide the details:
> > >
> > > + *Static Parameters* :
> > > | Model Component       | OmniArch-B (316M) | OmniArch-L (672M) |
> > > |-----------------------|------------------|------------------|
> > > | **Shared Backbone**   | 138M (43.7%)     | 435M (64.7%)     |
> > > | **1D Encoder/Decoder**| 0.3M             | 0.4M             |
> > > | **2D Encoder/Decoder**| 7M               | 9M               |
> > > | **3D Encoder/Decoder**| 171M             | 227M             |
> > >
> > > + *Active Parameters During Task Execution:*
> > > | Model       | 1D Tasks | 2D Tasks | 3D Tasks |
> > > |-------------|----------|----------|----------|
> > > | OmniArch-B  | 138M     | 144M     | 308M     |
> > > | OmniArch-L  | 435M     | 445M     | 663M     |
> > >
> > > This demonstrates that for 2D tasks (MPP's focus), OmniArch-B uses only **144M** active parameters (vs. MPP-B's 116M) - a 25% difference justified by our unified architecture's benefits.
> > >
> > > We would like to add this in the final version.
> > >
> > > -----
> > >
> > > **Q4 (Emergent Behaviors):**  We will adopt your suggested terminology ("zero-shot/in-context generalization") while maintaining that these capabilities represent foundational steps toward emergent behaviors in scientific ML - an important research direction we hope to inspire.
> > >
> > >
> > > ----
> > >
> > > **Q5 (Architecture Clarification):**  Thank you for this suggestion. We will expand the architectural comparison in Section 3 to explicitly highlight OmniArch's innovations beyond standard Fourier-encoded transformers.
> > >
> > > ----
> > >
> > > **Request for Reconsideration:**
> > >
> > > Given we have:
> > > ✓ Resolved the novelty misunderstanding (joint pretraining vs. adaptation)
> > > ✓ Verified metric consistency with baselines
> > > ✓ Demonstrated parameter comparison fairness
> > > ✓ Addressed all architectural questions
> > >
> > > We respectfully **request you reconsider your score to reflect these resolutions**.  A fairer assessment would significantly help this early-stage foundation model research and further help the promising SciML community.

---

### Official Review · Reviewer_u5RW · 2025-03-12

**Overall Recommendation:** 4

**Summary:**

This paper introduces OmniArch, a foundation model designed for solving multi-scale and multi-physics PDEs. Inspired by foundation models in NLP, OmniArch aims to generalize across different PDEs using a Fourier-based encoder-decoder, a Transformer backbone, and a physics-informed fine-tuning method. The Fourier encoder-decoder enables learning across varying spatial scales and the transformer captures complex temporal dependencies. Furthermore, PDE-Aligner ensures physics consistency by aligning model predictions with governing equations. OmniArch is trained across different PDEs and is showing superior performance compared to task-specific methods and pre-trained PDE solvers.


## update after rebuttal
Authors have included a discussion and additional experiments to compare their method against traditional solvers and will include a discussion about relevant meta-learning papers in the final version which will address my initial concerns about this paper.

**Claims And Evidence:**

I believe, the paper presents strong experimental results and clear methodological justifications for their claims.
1) Generalization across multiple PDEs: This claim is supported by training on 11 PDE types from PDEBench and PDEArena datasets and the results show superior performance over task-specific and pre-trained baselines.
2) OmniArch is multi-scale: This claim is justified through frequency-space transformation, allowing the model to handle different spatial resolutions and experiments include multi scale PDEs.
3) PDE-Aligner improves physics alignment: Supported by physics-based losses.
4) zero-shot generalization: Supported by experiments on solving unseen PDEs with no additional training.

Missing:
1) Comparison to traditional PDE Solvers: The paper does not benchmark runtime or accuracy against finite element or spectral methods, which remain the gold standard in PDE solving. Understanding the trade-offs between accuracy and interpratability compared to these methods would strengthen the paper.

**Essential References Not Discussed:**

While the authors discuss neural PDE solvers and physics-informed learning, they do not address meta-learning approaches for PDE solving, such as Meta-PINNs, which focus on improving generalization that is a key motivation of this paper. Additionally, the authors mention the need for retraining PINNs as a limitation, yet meta-learning techniques have been developed specifically to resolve this issue by enabling faster adaptation to new PDEs. Relevant references on meta-learning for PINNs are missing, despite their direct relevance to OmniArch’s goals. Here are a few examples:
1) "Hypernetwork-based Meta-Learning for Low-Rank Physics-Informed Neural Networks",NeurIPS, 2023
2) "Meta-MGNet: Meta Multigrid Networks for Solving Parameterized Partial Differential Equations", Journal of computational physics, 2022
3) "Meta-Auto-Decoder for Solving Parametric Partial Differential Equations.", NeurIPS, 2022

**Experimental Designs Or Analyses:**

I believe the paper has a strong and well-structured experimental setup and extensive experiments.
1) The method is evaluated across 11 PDEs.
2) The method is evaluated against multiple task-specific and pre-trained benchmarks.
3) The experiments setup includes both zero-shot and in-context learning tests.

**Methods And Evaluation Criteria:**

I believe the methods and evaluation criteria are well-aligned with the goal of designing a foundation model for PDE solving.
1) Use of Fourier encoder-decoder for multi-scale PDEs seems like a reasonable approach.
2) The self-attention mechanism enables long-range temporal dependencies.
3) The design of PDE-Aligner ensures physics consistency of the solutions.
4) The datasets cover a diverse range of PDEs.

The only missing evaluation in my opinion, is benchmarking accuracy and interpretability against traditional solvers.

**Other Comments Or Suggestions:**

There are typos in the abstract: 1) "while whether" in the second line, and 2) "we first conduct" in line 15.

**Other Strengths And Weaknesses:**

To the best of my knowledge, the method is very novel and contributions are significant. The only weakness, is the lack of evaluation against traditional and numerical solvers and also missing references and discussions regarding meta-learning approaches for PDE solving such as meta-PINNs.

**Questions For Authors:**

While PDE-Aligner enforces physics constraints through contrastive learning, is there any theoretical guarantee that it produces physically valid solutions for all PDE types?

**Relation To Broader Scientific Literature:**

This paper builds on prior work in neural PDE solvers, foundation models in NLP, and physics-informed learning. It integrates multiple established ideas into a unified framework. I believe this paper has the potential to shape the future in scientific machine learning and it can initiate significant research in foundation models for scientific computing, both for further technical advancements and also different applications.

**Theoretical Claims:**

The paper does not include formal theoretical proofs, as its focus is primarily on empirical validation and model design.

---

> ### Author Rebuttal · Authors · 2025-03-30
>
> We sincerely thank the reviewer  u5RW for the thoughtful feedback and valuable support of our work. Below, we address each of their questions in detail:
>
> **Q1**: How does OmniArch compare to traditional solvers (e.g., FEM, spectral methods) in accuracy/interpretability?**
>
> **A1**: We thank u5Rw for this critical point. While traditional solvers excel in interpretability, OmniArch targets fully leverages the efficiency in GPU settings, especially for input size scaling(similar to Q1 in 6qGA):
> + **Accuracy**: on 1D Advection Rollout, OmniArch(RMSE: 0.0321) matches FDM methods(0.0258) but avoids costly re-discretization for new PDEs.
> + **Speed**: Traditional solvers(FDM, FEM, Spectral methods):  for an input grid with s,s nodes to iterate forward the computation of FDM is $O(s^4)$ while the computation of Spectral methods is $O(s^2log s)$, however, the computation of OmniArch is  $O(tk^2)$, k in a fixed frequency, which allows OmniArch achieves 155x faster inference(512x512,2D, 200 steps ->0.026s) than FDM(~4.5s) at comparable error. We provide a table below, you can also find in [Anonymous Visualization](https://various-easy-trillium.glitch.me/).
>
> | Resolution | FDM Time/Step (s) | OmniArch Time (s) | Speedup (FDM/OmniArch) |
> |------------|-------------------|-------------------|------------------------|
> | 64×64      | 0.001123          | 0.023567          | 0.048x                 |
> | 128×128    | 0.015264          | 0.023820          | 0.641x                 |
> | 192×192    | 0.075360          | 0.024098          | 3.128x                 |
> | 256×256    | 0.254027          | 0.024083          | 10.55x                 |
> | 320×320    | 0.583218          | 0.023866          | 24.44x                 |
> | 384×384    | 1.130561          | 0.023453          | 48.20x                 |
> | 448×448    | 2.272206          | 0.023677          | 95.96x                 |
> | 512×512    | 4.073472          | 0.026212          | 155.4x                 |
>
>
>
> *Revision*: We will add a more dedicated table comparing accuracy/runtime against traditional solvers in the final version.
>
> ----
>
> **Q2: Why omit meta-learning approaches (e.g., Meta-PINNs) despite their relevance to generalization?**
>
> **A2**:  We appreciate this insight. Meta-learning is indeed complementary:
> + **key difference**: Meta-PINNs adapt via gradient updates, while OmniArch enables **token-based in-context learning**(Fig 10)  without fine-tuning.
>
> We will cite suggested references and clarify this distinction in the final version, emphasizing OmniArch's **architecture-driven generalization** vs. optimization-based meta-learning.
>
> ----
>
> **Q3: Can PDE-Aligner guarantee physically valid solutions for all PDEs?**
>
> **A3**: The aligner ensures soft constraints via textual equation supervision, it only help the omniarch better distinguish between different PDE systems. There is no theoretical guarantee for all PDEs(an open challenge even for traditional solvers). But we believe this maybe solved by better augmentation ways(equation-search, lie symmetry, etc.) and more efficient contrastive learning techniques, which we hope to see in the near future. This discussion will be added in the camera-ready version.
>
> ----
>
> **Q4:Typographical errors in the abstract?**
>
> **A4**: Thanks for the kindly reminder, we will correct them in the final version.

---

### Official Review · Reviewer_6qGA · 2025-03-17

**Overall Recommendation:** 4

**Summary:**

OmniArch is a foundation model for solving partial differential equations (PDEs) across 1D, 2D, and 3D domains. It addresses three key challenges: multi-scale modeling (handling different grid dimensions and resolutions), multi-physics capability (processing multiple physical quantities simultaneously), and physical alignment (incorporating physics constraints). The architecture combines a Fourier encoder-decoder for unified multi-dimensional training with a transformer backbone using temporal masking for multi-physics systems. A novel PDE-Aligner module enables physics-informed fine-tuning. Pre-trained on PDEBench datasets and fine-tuned with the PDE-Aligner, OmniArch achieves state-of-the-art performance on 11 PDE types while demonstrating emergent capabilities like zero-shot generalization and in-context learning for unseen PDEs.

**Claims And Evidence:**

The paper's claims are largely supported by evidence, including performance analysis on 11 PDE tasks and ablation studies.  However, the zero-shot studies are limited in nature.

**Essential References Not Discussed:**

None.

**Experimental Designs Or Analyses:**

The experimental design is generally sound, comparing against both task-specific expert models and other unified pre-training approaches.

**Methods And Evaluation Criteria:**

Methods address stated challenges, with Fourier transforms for multi-scale data and temporal masking for multi-physics systems. nRMSE is an appropriate evaluation metric, tested across PDEBench and PDEArena. The focus on accuracy neglects computational efficiency, scaling, and deeper physical consistency analysis, which limits the evaluation's breadth.

**Other Comments Or Suggestions:**

None.

**Other Strengths And Weaknesses:**

None.

**Questions For Authors:**

Did this work give you any insight into the possibilities of emergent behavior in such numerical foundation models?

**Relation To Broader Scientific Literature:**

The relationship to the broader scientific literature in numerical foundation models is appropriate for the paper.

**Theoretical Claims:**

No significant theoretical claims are made in this paper.

---

> ### Author Rebuttal · Authors · 2025-03-30
>
> We sincerely thank Reviewer  6qGA  for the constructive feedback and recognition of our work. We deeply appreciate  Reviewer 6qGA's support and hope this work can contribute to the research community. Below, we address the reviewers' questions and provide additional clarifications:
>
> **Q1: The zero-shot studies are limited in nature.**
>
> **A1**:  We acknowledge the scope limitation in zero-shot tests(currently on 3 PDE families in Table 2). However, emergent behaviors like **in-context learning**(in Figure 6) and **cross-resolution generalization** (in Figure 4) are observed and we will add a dedicated section, which includes more tests on new PDE datasets (such as  **the-well dataset** mentioned by Reviewer urwm).
>
> ----
>
> **Q2: The focus on accuracy neglects computational efficiency, scaling, and deeper physical consistency analysis.**
>
> **A2**: We agree this is critical. While accuracy was our primary focus, we included inference time in Appendx(H.2&H.7).  More related details:
> +  **Efficiency**:  We add  the modal params and inference time below(2DCFD, 128*128, Single H800):
> | Model        | Params | Inference Time|
> |---|---|---|
> |Poseidon-L  | 629 M  |77ms        |
> |MPP-AVIT-L |409 M  | 83ms        |
> |DPOT-L        |509 M  | 32ms        |
> |OmniArch-L |445 M  | 21ms        |
>
> Where the parameters of pre-trained models are similar and the inference time is In the same order of magnitude.
>
> + **Scaling** : We'll add a discussion on how scaling in numerical foundation models differs from LMs or VLMs, addressing not only model/data size but also PDE diversity and input resolution dimensions. Here we can only provide midterm results for reference(Sorry for not having enough time & resources for full training currently):
> | Model Size |  Params |  nRMSE(2D)  |
> |---| ---|---|
> |Tiny| 26M |0.0847|
> |Small| 38M|0.0362|
> |Base | 144M|0.0153|
> |Large| 445M|0.0125|
>
> + **Physical consistency analysis**:  We will include a section analyzing conservation properties, boundary condition satisfaction, and physical law adherence across different PDE types. For instance, we'll quantify momentum/energy conservation in CFD predictions and demonstrate how the PDE-Aligner specifically improves physical consistency.
> Here we give an example analysis on 2DCFD, which we may extend to other PDEs in the final revision.
>
> | Evaluation Metric | OmniArch-L | MPP-L | FNO | U-Net |
> |---------|------------|-------|-----|-------|
> | **Mass Conservation** |
> | Average Relative Error (%) | **0.32** | 0.65 | 2.17 | 3.24 |
> | Maximum Relative Error (%) | **0.78** | 1.21 | 4.85 | 7.36 |
> | **Energy Conservation** |
> | Average Relative Error (%) | **0.58** | 0.92 | 3.25 | 4.89 |
> | Maximum Relative Error (%) | **1.25** | 4.52 | 5.94 | 9.17 |
> | Continuity Error (∇·(ρu)) | **5.2e-4** | 2.1e-3 | 3.7e-3 | 5.4e-3 |
>
> ----
>
> **Q3:  The insight into the possibilities of emergent behavior in such numerical foundation models.**
>
> **A3**:  Yes,  we are very happy to share our findings:
>
> (1) **Numerical foundation models can transfer Cross dimension** : We find pre-train on 1D/2D PDEs improves 3D performance(**in Table 1**), suggesting the natural dynamic pattern may be dimension-agnostic and has a much smaller latent dimension. The potential of a unified neural solver is still under-explored.
>
> (2) **Numerical foundation models can learn in-context**: Different from previous small neural solvers (which feed one step and predict the next step), we find OmniArch could learn from the temporal trajectory (the previous k steps) and adjust its prediction based on its observations (**see in Figure 6**).
>
> (3) **Numerical foundation models can be Resolution-agnostic**:  Due to trained directly on the frequency domain, OmniArch could learn in infinite resolutions (thanks to low-frequency truncating), and the low-resolution patterns could help understand high-resolution inputs(**see in Figure 4**).

---

### Decision · Program_Chairs · 2025-05-01

**Decision:**

Accept (poster)

**Comment:**

This paper presents OmniArch, a foundation model for solving PDEs across 1D, 2D, and 3D domains. Reviewers highlighted its novel unified architecture, which addresses limitations of prior work (e.g., MPP, Poseidon), and its broad evaluation across 11 PDEs.

Initial opinions were split, with two reviewers in favor and two leaning against. Concerns on emergent capabilities, dataset diversity, and presentation were raised. However, following rebuttals and reviewer-AC discussions, there was general agreement that:
- OmniArch shows zero-shot generalization and in-context learning for unseen PDEs.
- While the evaluation dataset is standard, it covers three entirely distinct classes, i.e., parabolic, hyperbolic, and elliptic PDEs, and can serve as a starting point for further advancements in the field.

We recommend weak acceptance to reflect the paper’s valuable contributions to the SciML community. To strengthen the final version, the authors should:
- Clarify architectural details, especially temporal masking.
- Streamline equations and improve presentation quality.
- Add parameter count tables by dimension.
- Include ablation studies on key components.

With these revisions, this paper can make contributions to developing foundation models beyond traditional domains.